# TGFβ signalling is required to maintain pluripotency of human naïve pluripotent stem cells

Anna Osnato[1,2], Stephanie Brown[1,2], Christel Krueger[3], Simon Andrews[3], Amanda J Collier[4], Shota Nakanoh[1,2,5], Mariana Quiroga Londoño[1,2], Brandon T Wesley[1,2], Daniele Muraro[1,2,6], A Sophie Brumm[7], Kathy K Niakan[7,8], Ludovic Vallier[1,2†]*, Daniel Ortmann[1,2†]*, Peter J Rugg-Gunn[1,4,8†]*

[1]Wellcome–MRC Cambridge Stem Cell Institute, Jeffrey Cheah Biomedical Centre, University of Cambridge, Cambridge, United Kingdom; [2]Department of Surgery, University of Cambridge, Cambridge, United Kingdom; [3]Bioinformatics Group, The Babraham Institute, Cambridge, United Kingdom; [4]Epigenetics Programme, The Babraham Institute, Cambridge, United Kingdom; [5]Division of Embryology, National Institute for Basic Biology, Okazaki, Japan; [6]Wellcome Sanger Institute, Hinxton, Cambridge, United Kingdom; [7]Human Embryo and Stem Cell Laboratory, The Francis Crick Institute, London, United Kingdom; [8]Centre for Trophoblast Research, University of Cambridge, Cambridge, United Kingdom

**Abstract** The signalling pathways that maintain primed human pluripotent stem cells (hPSCs) have been well characterised, revealing a critical role for TGFβ/Activin/Nodal signalling. In contrast, the signalling requirements of naïve human pluripotency have not been fully established. Here, we demonstrate that TGFβ signalling is required to maintain naïve hPSCs. The downstream effector proteins – SMAD2/3 – bind common sites in naïve and primed hPSCs, including shared pluripotency genes. In naïve hPSCs, SMAD2/3 additionally bind to active regulatory regions near to naïve pluripotency genes. Inhibiting TGFβ signalling in naïve hPSCs causes the downregulation of SMAD2/3-target genes and pluripotency exit. Single-cell analyses reveal that naïve and primed hPSCs follow different transcriptional trajectories after inhibition of TGFβ signalling. Primed hPSCs differentiate into neuroectoderm cells, whereas naïve hPSCs transition into trophectoderm. These results establish that there is a continuum for TGFβ pathway function in human pluripotency spanning a developmental window from naïve to primed states.

*For correspondence:
lv225@cam.ac.uk (LV);
daniel.ortmann@gmail.com (DO);
peter.rugg-gunn@babraham.ac.uk
(PJR-G)

†These authors contributed equally to this work

Competing interests: The authors declare that no competing interests exist.

## Introduction

Human pluripotent stem cells (hPSCs) are grown in vitro as two broadly different states termed naïve and primed (*Davidson et al., 2015*; *Weinberger et al., 2016*). The two states diverge in their embryonic identity with primed hPSCs recapitulating post-implantation epiblast, and naïve hPSCs resembling pluripotent cells of pre-implantation embryos (*Rossant and Tam, 2017*; *Weinberger et al., 2016*). This difference has profound consequences on the cell's properties, including the epigenetic state and differentiation capacity (*Dong et al., 2019*). Naïve hPSCs have decreased DNA methylation levels, altered distribution of histone marks, and two active X-chromosomes, and they have a higher propensity to differentiate into extraembryonic tissues (*Castel et al., 2020*; *Cinkornpumin et al., 2020*; *Dong et al., 2020*; *Guo et al., 2021*; *Io et al., 2021*; *Linneberg-Agerholm et al., 2019*; *Pastor et al., 2016*; *Sahakyan et al., 2017*; *Takashima et al., 2014*; *Theunissen et al., 2016*; *Vallot et al., 2017*). On the other hand, primed hPSCs represent the last

stage before differentiation into the three definitive germ layers – ectoderm, mesoderm, and endoderm – from which the adult organs are derived (*Weinberger et al., 2016*).

Importantly, these pluripotent states are established by using specific and distinct culture conditions (*Taei et al., 2020*). Of particular interest, primed hPSCs rely on TGFβ/Activin/Nodal signalling to maintain their self-renewal and differentiation capacity (*James et al., 2005*; *Vallier et al., 2005*). Inhibition of these pathways or knock down of their effectors – SMAD2/3 – result in the rapid differentiation towards the neuroectoderm lineage (*Smith et al., 2008*). Conversely, an increased activity of these signalling pathways is necessary for endoderm differentiation (*D'Amour et al., 2005*; *Touboul et al., 2010*). The mechanisms behind these apparent divergent functions remain to be fully uncovered, but the capacity of SMAD2/3 to switch partners during differentiation is likely to play a key role (*Brown et al., 2011*). Of note, Epiblast Stem Cells (EpiSCs) derived from post-implantation mouse embryos also rely on TGFβ/Activin/Nodal signalling (*Brons et al., 2007*). Furthermore, genetic studies in the mouse have shown that Nodal signalling is necessary to block neuroectoderm differentiation and to maintain the expression of pluripotency markers in the post-implantation epiblast (*Camus et al., 2006*; *Mesnard et al., 2006*). Thus, the central role of TGFβ/Activin/Nodal in primed pluripotency seems to be evolutionary conserved and is important for normal development.

In contrast, the function and evolutionary conservation of TGFβ/Activin/Nodal signalling in pre-implantation embryos is less well understood. TGFβ/Activin/Nodal signalling does not have an essential role in forming the pre-implantation epiblast in mouse (*Brennan et al., 2001*; *Varlet et al., 1997*), whereas recent studies have suggested that the same pathway may be necessary for epiblast development in human blastocysts (*Blakeley et al., 2017*). The mechanistic basis for these observations are unclear. Moreover, it also remains to be established whether TGFβ signalling is required to maintain naïve hPSCs, which are the in vitro counterparts of pre-implantation epiblast cells. In general, naïve pluripotency is believed to be a steady state induced predominantly by blocking differentiation signals. However, the culture conditions vary between laboratories, although interestingly, most media that support naïve hPSCs contain exogenous TGFβ/Activin or a source of TGFβ provided by inactivated fibroblasts or Matrigel-coated substrates (*Bayerl et al., 2021*; *Chan et al., 2013*; *Gafni et al., 2013*; *Guo et al., 2016*; *Takashima et al., 2014*; *Theunissen et al., 2014*). Collectively, these observations suggest there could be an uncharacterised role for TGFβ/Activin/Nodal signalling specifically in the human naïve pluripotent state.

Here, we address this hypothesis by first establishing that TGFβ/Activin/Nodal signalling is active in naïve hPSCs. Using genome-wide analyses, we then show that SMAD2/3 is bound near to genes that characterise the naïve pluripotent state. Furthermore, loss of function experiments demonstrate that this signalling pathway is necessary to maintain the expression of key pluripotency genes, such as *NANOG*. We then perform single-cell RNA sequencing analyses on naïve and primed hPSCs that are undergoing differentiation following the inhibition of TGFβ/Activin/Nodal signalling. In these conditions, primed hPSCs rapidly decrease pluripotency markers and activate neuroectoderm genes, whereas naïve hPSCs induce trophectoderm markers. Importantly, these analyses also suggest that SMAD2/3 directly maintains an important part of the transcriptional network characterising the naïve state. Taken together, these results show that TGFβ/Activin/Nodal signalling is necessary to maintain the pluripotent state of naïve hPSCs through directly sustaining the expression of key pluripotency genes. These new insights suggest that the function of TGFβ/Activin/Nodal signalling in human pluripotency extends to earlier stages of development than previously anticipated, thereby underlying a key species divergence that could facilitate the identification and the isolation of pluripotent states in vitro.

## Results

### TGFβ signalling pathway is active in human naïve pluripotent cells

To assess whether the key effectors of the TGFβ signalling pathway are expressed in naïve hPSCs and to evaluate the cell heterogeneity in their expression (*Figure 1a*; *Figure 1—figure supplement 1a,b*), we performed single cell transcriptomic analysis (scRNA-seq) in naïve and primed hPSCs (*Figure 1b*; *Figure 1—figure supplement 1c*). As expected, naïve and primed hPSCs clustered separately based on their transcriptomes. All cells expressed pan-pluripotency genes, such as *POU5F1* (also known as *OCT4*), *NANOG* and *SOX2*, however, naïve cells uniquely expressed known naïve cell

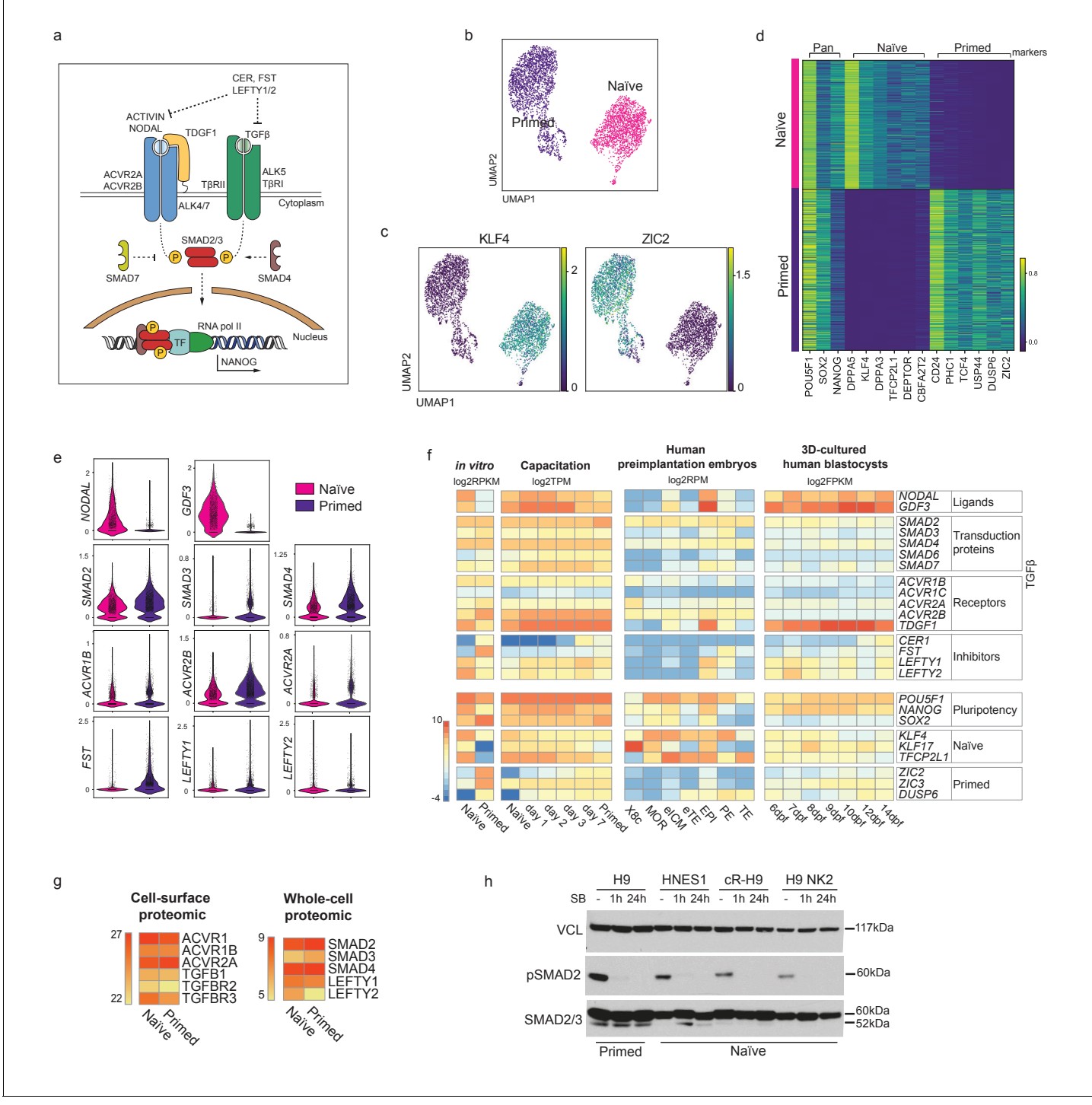

**Figure 1.** TGFβ signalling pathway is active in human naïve pluripotent stem cells. (**a**) Overview of the TGFβ signalling pathway. Extracellular ligands ACTIVIN and NODAL bind to type I (ACVR2A/2B) and type II transmembrane receptors (ALK4/7), and TGFβ binds to TβRI and TβRII/ALK5. NODAL requires the additional transmembrane co-receptor TDGF1 (CRIPTO1). The activated receptor complex phosphorylates the linker region of SMAD2 and SMAD3, which enter the nucleus in complex with SMAD4. They act as transcriptional regulators and induce or repress the transcription of their target loci by recruiting other transcription factors (TF) and epigenetic modifiers. Several negative regulators of the signalling pathway are also shown: LEFTY1/2 block the signalling pathway by binding to the receptors; Cerberus (CER) and Follistatin (FST) block the ligands; SMAD7 inhibits the SMAD2/3 complex. (**b**) 10X RNA-seq data of naïve and primed hPSCs represented on a UMAP plot. (**c**) UMAP visualisation of naïve and primed hPSCs reporting the relative expression of respective pluripotent state markers, *KLF4* and *ZIC2*. (**d**) Heatmap reporting the expression values of selected naïve and primed marker genes divided in pan-pluripotency markers, and naïve- and primed-specific markers within the top 250 differentially expressed genes. (**e**)

*Figure 1 continued on next page*

*Figure 1 continued*

Violin plots of the 10X RNA-seq data comparing the transcript expression of TGFβ effectors in naïve and primed hPSCs. (f) Heatmap summarising the transcript expression of TGFβ effectors and pluripotency genes. RNA-seq datasets shown are: in vitro-cultured naïve and primed hPSCs (*Collier et al., 2017*), hPSCs undergoing naïve to primed state capacitation (*Rostovskaya et al., 2019*), human pre-implantation embryos (*Petropoulos et al., 2016*), and epiblast cells within a 3D human blastocyst culture system (*Xiang et al., 2020*). X8c: 8-cell stage; MOR: morula; eICM: early-ICM; eTE: early-trophectoderm; EPI: epiblast; PE: primitive endoderm; TE: trophectoderm. Dpf: days post-fertilisation. (g) Heatmaps summarising protein abundance levels determined by cell-surface proteomics (*Wojdyla et al., 2020*) and whole cell proteomics (*Di Stefano et al., 2018*) for TGFβ effectors in naïve and primed hPSCs. (h) Western blot analysis of TGFβ signalling pathway activation in H9 primed hPSCs (cultured in E8 medium) and in three naïve hPSC lines cultured in t2iLGö medium: embryo-derived HNES1, chemically reset cR-H9, and transgene-reset H9 NK2. Blots show SMAD2 phosphorylation signal (pSMAD2-Ser465/Ser467) and total SMAD2/3 levels in normal conditions (-), and following 1 hr and 24 hr of SB-431542 supplementation to their culture media. Vinculin (VCL) used as a loading control.

The online version of this article includes the following source data and figure supplement(s) for figure 1:

**Source data 1.** Full uncropped western blot from *Figure 1h* and *Figure 1—figure supplement 1e* reporting TGFβ pathway activation in primed H9 cells (cultured in E8 medium) and in three naïve hPSC lines cultured in t2iLGö medium: embryo-derived HNES1, chemically reset cR-H9, and transgene-reset H9 NK2 naïve cells.

**Figure supplement 1.** Validation of naïve and primed hPSCs and TGFβ signalling pathway activation.

**Figure supplement 1—source data 1.** Numerical data that are represented in *Figure 1—figure supplement 1*.

markers, such as *DPPA5* and *KLF4,* and primed cells expressed *CD24* and *ZIC2* (*Figure 1c,d*; *Figure 1—figure supplement 1c*). In addition, differential expression analysis confirmed the specific expression of naïve hPSCs genes, such as *KLF4*, *DPPA3*, and *TFCP2L1*, and primed hPSCs factors including *DUSP6*, *ZIC2*, and *TCF4* (*Figure 1d*). Importantly, we found that most TGFβ pathway effectors, such as Activin receptors (*ACVR*s) and *SMAD2-4*, are expressed at similar levels in both pluripotent cell types (*Figure 1e*). Interestingly, several components, including *NODAL* and *GDF3*, have higher expression levels in naïve compared to primed hPSCs (*Figure 1e*).

We next examined RNA-seq datasets that covered different stages of human pluripotency in stem cell lines and in embryos (*Figure 1f*). We first compared naïve and primed hPSCs (*Collier et al., 2017*) and, consistent with our scRNA-seq data, we found that most ligands, transduction proteins and receptors of the TGFβ pathway are expressed at similar levels in the two cell types (*Figure 1f*). Higher expression of the TGFβ ligands *NODAL* and *GDF3* and the co-receptor *TDGF1* was again detected in naïve hPSCs. Interestingly, the expression of pathway inhibitors differed, whereby *LEFTY1* and *LEFTY2* were higher in naïve hPSCs, whereas *CER1* and *FST* were higher in primed hPSCs. We then looked at gene expression changes that occur during the process of capacitation, because the transition from naïve to primed hPSCs recapitulates pre- to post-implantation epiblast cell development (*Rostovskaya et al., 2019*). We found that most of the effectors of the TGFβ pathway are expressed throughout the entire developmental series, and also confirmed that *NODAL* and *GDF3* are expressed at higher levels in the early stages (*Figure 1f*).

To examine transcriptional events directly in human embryos, we next looked at scRNA-seq data in human pre-implantation embryos from day 3 to day 7 (*Petropoulos et al., 2016*). Low level expression of most TGFβ pathway effectors was detected in the early inner cell mass (ICM), and their expression increased substantially in the pre-implantation epiblast (EPI). In particular, *NODAL* and *GDF3* are highly expressed in EPI at this stage, similar to the transcriptional patterns in naïve hPSCs (*Figure 1f*). However, in contrast to EPI, most pathway components are undetectable in trophectoderm (TE and early TE), and are expressed at low levels in primitive endoderm (PE) (*Figure 1f*). These observations were extended by examining the expression of TGFβ pathway genes in a blastocyst-culture system that recapitulates EPI development from pre-implantation to early gastrulation (*Xiang et al., 2020*). Here, in EPI cells at 6 days post-fertilisation, *NODAL*, *GDF3* and the NODAL co-receptor *TDGF1* are highly expressed, in line with the EPI stage from the Petropoulos et al. dataset, and the high expression of these genes is sustained in all EPI cells over the following eight days of development (*Figure 1f*). Taken together, these results show that most ligands, transduction proteins and receptors of the TGFβ pathway are expressed at similar levels in naïve and primed hPSCs, and that this expression pattern across pluripotent states is also observed in human embryos cultured in vitro.

To further confirm these observations at the protein level, we examined cell-surface proteomic (*Wojdyla et al., 2020*) and whole-cell proteomic (*Di Stefano et al., 2018*) data in naïve and primed

hPSCs. This revealed that most Activin/TGFβ receptors and downstream effectors of the pathways are expressed at very similar levels in the two cell types (*Figure 1g*; *Figure 1—figure supplement 1d*). Finally, to directly assess TGFβ pathway activation, we performed western blot analysis and found that phospho-SMAD2 (pSMAD2), the activated form of SMAD2, is detectable in multiple embryo-derived and reprogrammed naïve hPSCs lines, and at comparable levels to primed cells (*Figure 1h*; *Figure 1—figure supplement 1e,f*). The phosphorylation signal was rapidly diminished following the treatment of the cells with SB-431542 (SB), a potent and selective inhibitor that blocks TGFβ/Activin receptors ALK5, ALK4, and ALK7 (*Inman et al., 2002*; *Figure 1h*; *Figure 1—figure supplement 1e,f*). Taken together, these results establish that the TGFβ signalling pathway is active in naïve hPSCs. Because primed hPSCs rely on this pathway to maintain pluripotency, our findings raise the possibility that naïve hPSCs might also require TGFβ signalling to sustain their undifferentiated state.

## SMAD2/3 binding is enriched at active enhancers in human naïve cells

Having established that the TGFβ signalling pathway is active in naïve hPSCs, we next profiled the genome-wide occupancy of the main downstream effectors – SMAD2/3 – using chromatin immuno-precipitation combined with genome-wide sequencing (ChIP-seq) in naïve and primed hPSCs. This analysis revealed that SMAD2/3 binding is enriched in naïve cells to a similar degree as in primed cells, as shown by independent peak calling in the two cell types (*Figure 2a*; *Figure 2—figure supplement 1a*). Here, we observed regions bound by SMAD2/3 in both cell types, and also a substantial number of loci that appear to have cell-type-specific binding. Importantly, canonical target genes, such as *LEFTY1/2*, *NODAL*, *NANOG*, and *SMAD7*, were bound by SMAD2/3 in both cell types (*Figure 2b*; *Figure 2—figure supplement 1b*), suggesting that TGFβ is active and it signals through the canonical cascade in both naïve and primed hPSCs.

In addition to the shared targets, differential binding analyses revealed over 2000 SMAD2/3-bound sites that differed between the two cell types (*Figure 2c,d*; *Figure 2—figure supplement 1c, d*). Excitingly, further examination of these differential sites revealed that in naïve hPSCs SMAD2/3 uniquely bound near to naïve-specific pluripotency genes including *DNMT3L*, *TFAP2C*, *CBFA2T2*, *KLF4,* and *CDK19* (*Figure 2e*; *Figure 2—figure supplement 1d,e*). Interestingly, these sites often overlapped with accessible chromatin regions and H3K27ac marks, which are signatures that are associated with active enhancers (*Heintzman et al., 2009*; *Figure 2e*; *Figure 2—figure supplement 1e*). In contrast, primed-specific SMAD2/3 sites were located near to genes that regulate mesendoderm differentiation, such as *TBXT*, *EOMES*, and *GATA4*, or primed-state pluripotency, such as *OTX2* (*Figure 2e*; *Figure 2—figure supplement 1e*). These sites correspond mostly to accessible chromatin and to regions marked by H3K4me3 and H3K27me3 signals, which typically mark the promoters of developmental genes (*Azuara et al., 2006*; *Bernstein et al., 2006*; *Heintzman et al., 2009*; *Figure 2e*; *Figure 2—figure supplement 1e*). These findings are supported by global analysis using ChromHMM-based chromatin state annotations (*Chovanec et al., 2021*), where we found that most SMAD2/3 peaks are indeed within active chromatin regions, consisting mainly of gene promoters and enhancers (*Figure 2f*). Interestingly, naïve-specific SMAD2/3 peaks are slightly more enriched at active enhancers compared to primed-specific peaks (30.6% vs 21.4%), and primed-specific SMAD2/3 peaks are instead more enriched at promoters (46% vs 26.5%) (*Figure 2f*; *Figure 2—figure supplement 1f*).

There are widespread differences in enhancer activity between naïve and primed hPSCs (*Barakat et al., 2018*; *Battle et al., 2019*; *Chovanec et al., 2021*) and so to determine how changes in SMAD2/3 occupancy tracks with enhancer status we compared chromatin marks at naïve-specific SMAD2/3 sites between the two cell types. The vast majority of sites that lose SMAD2/3 occupancy in primed hPSCs also show a strong reduction in chromatin accessibility and H3K27ac/H3K4me1 signals, which suggests that SMAD2/3-bound enhancers are decommissioned in primed hPSCs (*Figure 2g*). Chromatin marks that denote promoters and heterochromatin regions are generally low at naïve-specific SMAD2/3 sites and are largely unchanged in primed hPSCs, further reinforcing the connection between SMAD2/3 occupancy and active enhancers in naïve hPSCs (*Figure 2h*).

To obtain a more complete view of the pluripotency transcriptional network, we also overlapped SMAD2/3 peaks in naïve cells with OCT4, SOX2, and NANOG (OSN) binding (*Chovanec et al., 2021*). We found that OSN signals were strongly reduced at naïve-specific SMAD2/3 sites in primed hPSCs, confirming the integration of SMAD2/3 within the naïve transcription factor network

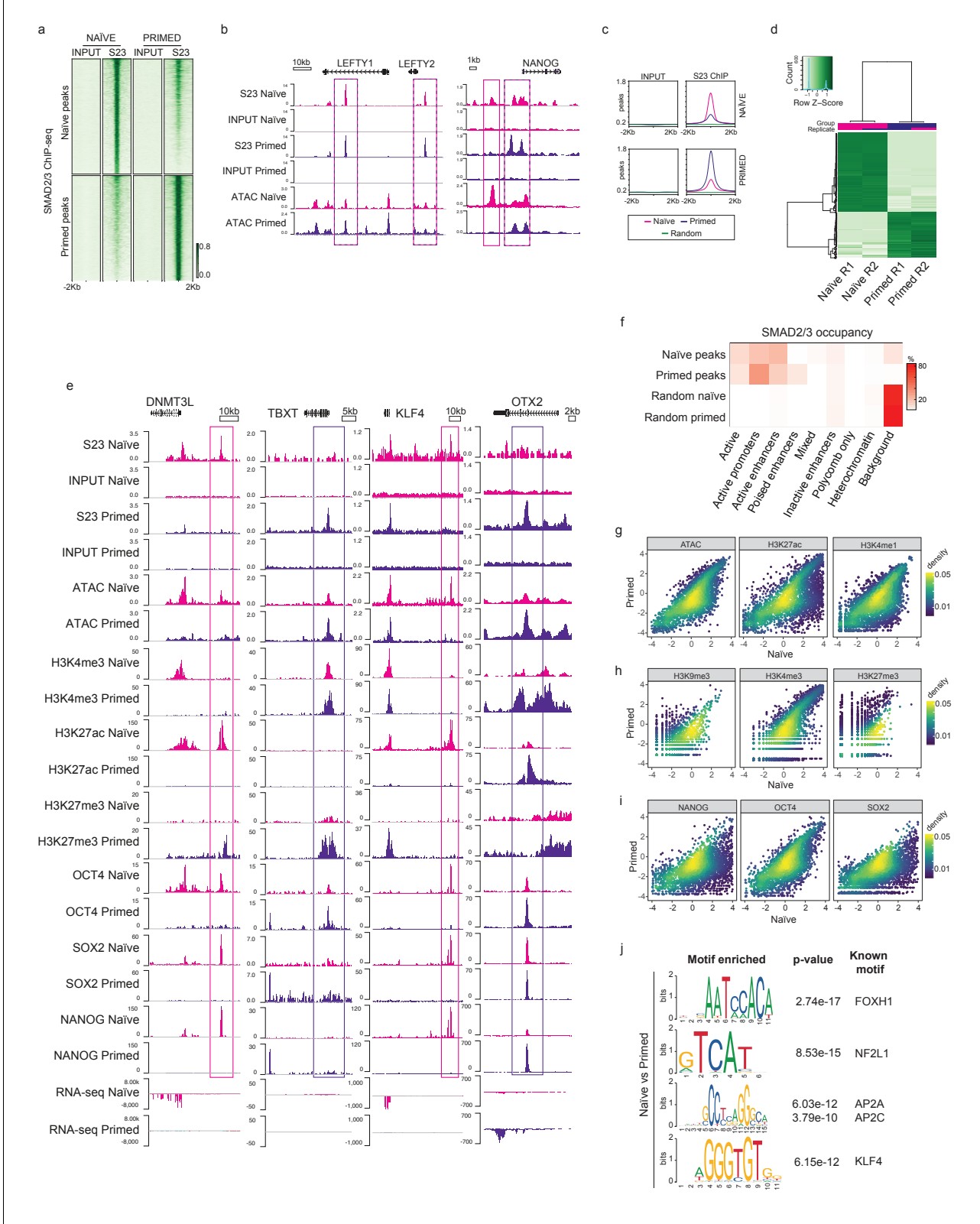

**Figure 2.** SMAD2/3 binds to chromatin at common and pluripotent state-specific sites. (**a**) Heatmap displaying normalised SMAD2/3 (S23) ChIP-seq reads ±2 kb from the centre of SMAD2/3-bound peaks that were independently defined in naïve (H9 NK2 line cultured in t2iLGö medium) and primed (H9 line cultured in E8 medium) hPSCs; two biological replicates per cell line. Top panel shows the regions identified as SMAD2/3-bound peaks in naïve cells; lower panel shows SMAD2/3-bound peaks in primed cells. (**b**) Genome browser tracks reporting SMAD2/3 (S23) binding (this study) and

*Figure 2 continued on next page*

Figure 2 continued

chromatin accessibility (ATAC-seq; *Pastor et al., 2018*) at the *LEFTY1/2* and *NANOG* loci in naïve and primed hPSCs. Input tracks are shown as controls. (c) Normalised average meta-plots of SMAD2/3 (S23) ChIP signal ±2 kb from the centre of the peaks in naïve and primed hPSCs, compared to a randomly-selected subset of regions. (d) Heatmap displaying regions that are differentially bound by SMAD2/3 in naïve and primed hPSCs in two biological replicates (R1 and R2). (e) Genome browser tracks reporting expression (RNA-seq), chromatin accessibility (ATAC-seq), and ChIP-seq datasets of SMAD2/3 (S23), histone marks for enhancers (H3K27ac) and promoters (H3K4me3, H3K27me3), and transcription factors (OCT4, SOX2, NANOG) at the *DNMT3L, TBXT, KLF4, OTX2* loci. Input tracks are shown as controls. The following data sets are shown: ATAC-seq (*Pastor et al., 2018*); H3K4me3 (*Theunissen et al., 2014*); H3K4me1 (*Chovanec et al., 2021*; *Gifford et al., 2013*); H3K27me3 (*Theunissen et al., 2014*); H3K27ac (*Ji et al., 2016*); OCT4 (*Ji et al., 2016*); SOX2 (*Chovanec et al., 2021*); NANOG (*Chovanec et al., 2021*; *Gifford et al., 2013*), and RNA-seq (*Takashima et al., 2014*). (f) Heatmap showing the frequency of SMAD2/3 peak centre locations with respect to ChromHMM states in naïve and primed hPSCs (*Chovanec et al., 2021*). SMAD2/3 peaks in naïve and primed hPSCs were annotated with their respective ChromHMM states. The annotations associated with the randomly-selected control regions reflect the overall genomic representation of chromatin states. (g-i) Density coloured scatter plots showing indicated ChIP-seq and ATAC-seq values (log$_2$ RPM) in naïve versus primed hPSCs. Each dot corresponds to one naïve-specific SMAD2/3 peak. (j) Differential motif enrichment reporting the top four motifs (ranked by p-value) at SMAD2/3-binding sites in naïve hPSCs that are enriched compared to motifs identified at SMAD2/3-binding sites in primed hPSCs.

The online version of this article includes the following figure supplement(s) for figure 2:

**Figure supplement 1.** SMAD2/3 binds to chromatin at common and state-specific sites.

(*Figure 2i*). Importantly, regions bound by SMAD2/3 and OSN overlapped with state-specific enhancers that are marked by open chromatin and H3K27ac, as shown for the *KLF4* and *DNMT3L* loci in naïve hPSCs, and for *OTX2* and *TBXT* in primed hPSCs (*Figure 2e*). Finally, to further characterise the differentially bound loci, we performed differential motif enrichment to investigate whether different binding partners might regulate SMAD2/3 binding in naïve and primed cells. Interestingly, motifs that are relatively enriched at SMAD2/3 sites in naïve compared to primed cells included NF2L1 (also known as NRF1), TFAP2A/C, KLF4 and FOXH1 (*Figure 2j*).

Altogether, these data suggest that SMAD2/3, the main effector of TGFβ pathway, is integrated in the naïve pluripotency network by targeting OSN-bound active enhancers that are in close proximity to key regulators of naïve pluripotency.

## Inhibiting TGFβ signalling induces loss of pluripotency in human naïve cells

After establishing that the TGFβ signalling pathway could maintain directly the transcriptional network characterising human pluripotency, spanning from naïve to primed states, we next examined whether the pathway is functionally required to sustain naïve hPSCs in an undifferentiated state. We first measured the transcriptional changes that occurred in response to SB-mediated loss of pSMAD2 and inhibition of the TGFβ pathway (*Figure 3a*; *Figure 3—figure supplement 1a,b*). After only 2 hours of SB treatment (t2iLGö medium supplemented with SB), naïve hPSCs showed a significant reduction in the expression of the pluripotency gene *NANOG*, which is a short time frame that is consistent with *NANOG* being a direct target of SMAD2/3 signalling (*Vallier et al., 2009*; *Xu et al., 2008*; *Figure 3a*; *Figure 3—figure supplement 1a*). Other canonical downstream target genes, such as *LEFTY1/2* and *SMAD7*, were also strongly downregulated and their expression was completely abolished after 24 hr in the case of *LEFTY1/2*. Excitingly, naïve pluripotency marker genes that are bound by SMAD2/3 including *DPPA3, DPPA5, KLF4,* and *DNMT3L* were also downregulated following SB treatment, indicating that the naïve state is disrupted in these conditions (*Figure 3a*; *Figure 3—figure supplement 1a*). These results were independently validated by depleting SMAD2/3 expression using the OPTiKD system (*Bertero et al., 2016*). Here, we generated stable naïve hPSCs with tetracycline (TET) inducible co-expression of shRNAs that target *SMAD2* and *SMAD3* transcripts (*Figure 3b*). Treating these cells with TET induced the rapid loss of *SMAD2/3* mRNA (*Figure 3c*), and a concomitant and significant downregulation in the expression of SMAD2/3 target genes, such as *LEFTY2, NODAL,* and *NANOG* (*Figure 3c*). We also detected a significant decrease in *POU5F1* expression following SMAD2/3 knockdown and after SB treatment, suggesting that naïve hPSCs are destabilised and are exiting the pluripotent state (*Figure 3a,c*).

Interestingly, adding SB to naïve culture media also induced a change in cell morphology whereby naïve hPSCs lost their typical dome-shaped morphology after 3 to 5 days, and this was accompanied by the appearance of flat colonies that gradually took over the culture (*Figure 3d*;

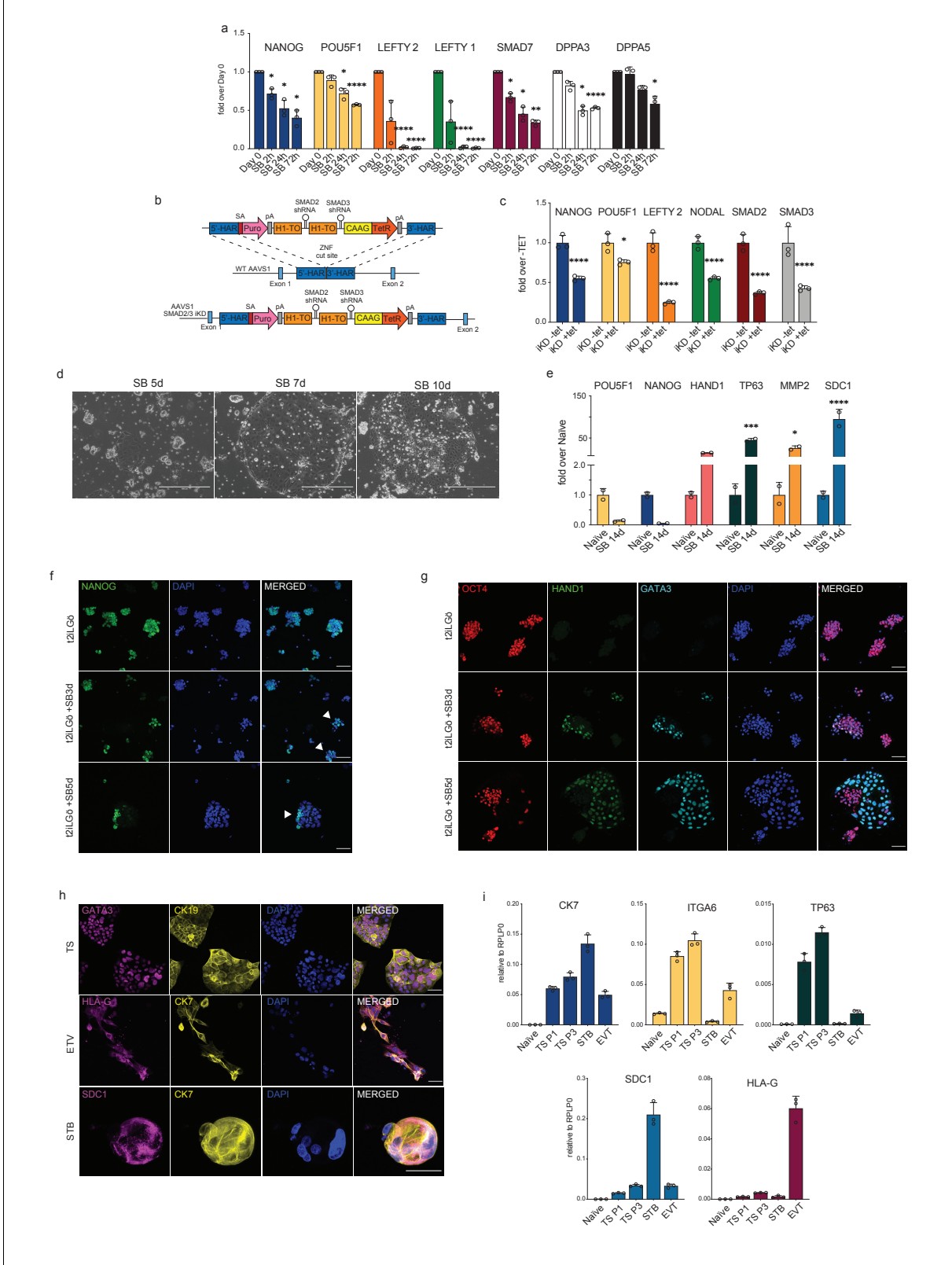

**Figure 3.** Inhibiting TGFβ signalling induces loss of pluripotency in naïve hPSCs. (**a**) RT-qPCR expression analysis of pluripotency-associated genes and TGFβ-associated genes in naïve hPSCs (H9 NK2 line) following SB-431542 treatment (t2iLGö + SB). Expression levels are shown as fold changes relative to day 0. (**b**) Schematic showing the integration of a single-step optimised inducible knock-down targeting construct into the *AAVS1* locus of H9 hPSCs, enabling the expression of SMAD2 and SMAD3 short hairpin RNAs (shRNAs) under the control of a tetracycline inducible promoter. ZFN: zinc-finger

*Figure 3 continued on next page*

Figure 3 continued

nucleases; 5'-HAR/3'-HAR: upstream/downstream homology arm; H1-TO: Tetracycline-inducible H1 Pol III promoter carrying one tet operon after the TATA box; CAAG: CMV early enhancer, chicken β-actin and rabbit β-globin hybrid promoter; TetR: Tetracycline-sensitive repressor protein; SA: splice acceptor; Puro, Puromycin resistance; pA, polyadenylation signal. Schematic adapted from *Bertero et al., 2016*. (c) RT-qPCR analysis of gene expression levels in SMAD2/3 inducible knock-down (iKD) H9 naïve hPSCs following 5 days of tetracycline (tet) treatment. Expression levels are shown for each gene as fold change relative to iKD -tet. Cells were cultured in t2iLGö medium. (d) Phase contrast pictures of H9 NK2 naïve hPSCs after 5, 7, and 10 days of SB treatment in t2iLGö medium. Scale bars: 400 µm. (e) RT-qPCR analysis of trophoblast (*HAND1*, *TP63*, *MMP2*, and *SDC1*) and pluripotency (*POU5F1*, *NANOG*) gene expression levels in naïve hPSCs following long-term (14 days) SB treatment in t2iLGö medium. Expression levels are shown as fold changes relative to day 0 samples, n = two biological replicates. (f) Immunofluorescence microscopy showing the downregulation of NANOG (green) in naïve hPSCs following 3 and 5 days of SB treatment. DAPI signal in blue. White arrowheads indicate colonies displaying heterogeneous expression of NANOG. Scale bars: 50 µm. (g) Immunofluorescence microscopy for OCT4 (red), HAND1 (green), GATA3 (cyan), and DAPI (blue) in naïve hPSCs following 3 and 5 days of SB treatment in t2iLGö medium. Scale bars: 50 µm. (h) Immunofluorescence microscopy for GATA3, HLA-G, SDC1 (magenta), CK19 and CK7 (yellow), and DAPI (blue) in naïve-derived trophoblast stem cells (TS), extravillous trophoblast (EVT), and syncytiotrophoblast (STB). Scale bars: 50 µm. (i) RT-qPCR analysis of gene expression levels in naïve-derived trophoblast stem cells (TS), extravillous trophoblast (EVT) and syncytiotrophoblast (STB) compared to undifferentiated naïve hPSCs. Expression levels are shown for each gene relative to the housekeeping gene *RPLP0*. RT-qPCR data show the mean ± SD of three biological replicates (unless specified otherwise) and were compared to their relative control using an ANOVA with Tukey's or Šídák's multiple comparisons test (*p ≤ 0.05, **p≤ 0.01, ***p≤ 0.001, ****p≤ 0.0001).
The online version of this article includes the following source data and figure supplement(s) for figure 3:

**Source data 1.** Numerical data that are represented in *Figure 3*.
**Figure supplement 1.** TGFβ signalling inhibition induces loss of pluripotency in different naïve hPSCs.
**Figure supplement 1—source data 1.** Full uncropped western blot from *Figure 3—figure supplement 1b* reporting TGFβ pathway activation in H9 NK2 naïve cells through the phosphorylation of SMAD2 (pSMAD2) and also total SMAD2/3 in normal conditions (-), after 1 hr and 2 hr of fresh media change (t2iLGö), and following 1 hr and 24 hr of SB treatment (t2iLGö+SB).

*Figure 3—figure supplement 1c*). This striking phenotypic change was confirmed in a second naïve hPSCs line (*Figure 3—figure supplement 1d*). Intriguingly, the morphology of these flat colonies resembles human trophoblast cells (*Okae et al., 2018*). To further investigate this, we grew naïve hPSCs for 14 days in the presence of SB and then examined the expression of trophoblast marker genes (*Figure 3e*). We found there was a strong upregulation in the expression of the trophecto-derm marker *HAND1* and also of *TP63*, *MMP2*, and *SDC1* that mark cytotrophoblast (CTB), extravil-lous trophoblast (EVT) and syncytiotrophoblast (STB) cell types, respectively (*Figure 3e*). These results were further supported by the clear reduction in NANOG protein expression following 3–5 days of treating naïve hPSCs with SB, in correspondence with the exit from naïve pluripotency and the appearance of the trophoblast-like colonies (*Figure 3f*). NANOG downregulation together with the appearance of trophoblast-like colonies was also observed in a second naïve cell line upon SB treatment (*Figure 3—figure supplement 1e*). Importantly, the flat cell colonies also expressed typi-cal trophoblast-associated proteins – GATA3 and HAND1 (*Figure 3g*; *Figure 3—figure supplement 1f,g*).

To further characterise these cells and to investigate their ability to differentiate into trophoblast derivatives, we cultured naïve hPSCs in the presence of SB for 5 days and then transferred the cells into trophoblast stem cell (TSC) media (*Dong et al., 2020*; *Okae et al., 2018*). Although the cell population initially appeared heterogeneous, following exposure to TSC conditions the cells rapidly and uniformly acquired a homogeneous TSC-like morphology. The cells expressed TSC markers, such as GATA3 and CK19 (*Figure 3h*) and *CK7*, *ITGA6*, and *TP63* (*Figure 3i*), and could be passaged and maintained in these conditions with stable growth and morphology. Naïve-derived TSCs were then induced to differentiate by switching the cells to STB and EVT media (*Dong et al., 2020*). This led to the downregulation of TSC genes and the upregulation of STB and EVT markers, such as SDC1 and HLA-G, respectively (*Figure 3h,i*).

Taken together, these results show that blocking TGFβ signalling in naïve hPSCs rapidly destabil-ises the pluripotency network and allows the cells to undergo differentiation toward trophoblast-like cells, including those that can give rise to multipotent, proliferative TSCs.

## Single-cell transcriptional analysis reveals a trophoblast-like population arising in response to TGFβ inhibition in human naïve cells

We next sought to investigate the processes in which TGFβ pathway inhibition drives naïve hPSCs out of their pluripotent state and towards a trophoblast phenotype. Following SB treatment, we

observed that the early-stage cultures contained a heterogeneous mixture of cell morphologies that included naïve-like colonies and the flat, TSC-like colonies described above (*Figure 3d*; *Figure 3—figure supplement 1c,d*). The proportion of NANOG-positive cells declined following SB treatment, with variable expression within individual colonies (*Figure 3f*). We also observed heterogeneous colonies that contained cells expressing the pluripotency marker OCT4 and TSC-like markers HAND1/GATA3 (*Figure 3g*). Because the population heterogeneity could mask important changes in cell phenotype, we used scRNA-seq to examine the effect of TGFβ inhibition over 7 days of SB treatment in naïve hPSCs (*Figure 4a*). In addition, to better characterise the divergent developmental potential between different human pluripotent states, we compared this response to the response when primed hPSCs were treated with SB. Our aim was to investigate the trajectory of naïve hPSCs moving into a putative TSC-like population, in contrast with the neuroectodermal differentiation that is induced in primed hPSCs when TGFβ is inhibited (*Vallier et al., 2009*).

In both cell types, there was a clear transcriptional trajectory moving from day 0 to day 7 of SB treatment (*Figure 4b*; *Figure 4—figure supplement 1a*). Importantly, there was little overlap in their trajectories (*Figure 4c*), confirming that the inhibition of TGFβ signalling in these two different developmental stages results in divergent differentiation processes. Louvain clustering of the combined datasets also showed separated clusters in the naïve and primed time course samples (*Figure 4—figure supplement 1b*). Specifically, TGFβ inhibition in naïve hPSCs induced the expression of TSC-like markers, such as *HAND1*, *GATA2*, and *GATA3*, whereas inhibition in primed hPSCs induced neuroectoderm markers, such as *SOX10*, *PAX6*, and *LEF1* (*Figure 4d*; *Figure 4—figure supplement 1c*). Interestingly, Louvain clustering of the naïve cell dataset initially follows the day 0 (Cluster A) and day 1 (Cluster B) timepoints and then resolves the mixed population at days 3, 5, and 7 into three separate clusters (C, D, and E) (*Figure 4e*; *Figure 4—figure supplement 1d*). This analysis suggests that the mixed population is formed from an early differentiating population (cluster C), a transition population (cluster D), and a later-stage differentiated population (cluster E), thereby confirming a stepwise process marked by different intermediate stages.

Examining individual genes revealed the dynamics of the differentiation trajectory. Pan-pluripotency and naïve-specific genes showed a gradient in their expression patterns, starting from high expression in cluster A, diminishing levels in clusters B and C, then largely absent in clusters D and E (*Figure 4f,g*). In contrast, trophoblast genes become activated in clusters C, D, and E, with *CDX2*, *HAND1* and *GATA3* marking early, transition and late-stage differentiating populations, respectively (*Figure 4f,g*). *NODAL* and *LEFTY1* are expressed predominantly in cluster A and were rapidly downregulated already in cluster B (*Figure 4f,g*), and other TGFβ pathway genes, such as *GDF3* and *TDGF1*, are fully downregulated when cells start transitioning towards cluster D. These results confirm the effective pathway inhibition and also that blocking TGFβ signalling allows trophectoderm differentiation.

To better characterise the Louvain clusters, we examined the top 25 genes that are differentially expressed in each cluster compared to all other clusters (*Figure 4g*; *Figure 4—figure supplement 1e,f*). Differentially expressed genes that are associated with cluster A, which corresponds largely to cells at day 0, include *NANOG* and *SUSD2* (*Bredenkamp et al., 2019a*; *Wojdyla et al., 2020*) in addition to the TGFβ ligand *GDF3* and receptor *TDFG1*. Interestingly, the SMAD2/3-cofactor *FOXH1* was also identified in this category and this is consistent with our prior motif analysis of the SMAD2/3 ChIP-seq data that identified FOXH1 as a putative interactor of SMAD2/3 specifically in naïve cells (*Figure 2j*). Genes that are differentially expressed in cluster B are enriched for metallothioneins, such as MT1/2 s, which affect cell respiration, in addition to mitochondrial genes – *SLIRP* and *MTNDL4* – and the glucose pyrophosphorylase *UGP2*, suggesting that an initial response to TGFβ inhibition could involve a metabolic switch (*Mathieu and Ruohola-Baker, 2017*). Cells in cluster C still express pluripotency markers, such as *POU5F1* and *DPPA5*, and have upregulated the non-coding RNAs *MEG3* and *MEG8*. Cluster D clearly marks a transition population towards TSC-like cells, with the expression of *CDX1* and *CDX2*, keratins (*KRT8*, *KRT18*), and *MARCKS*, *FABP5*, and *EZR* (*Cambuli et al., 2014*; *Ralston et al., 2010*). Cluster E includes keratins (*KRT8*, *KRT18*, *KRT19*), several main regulators of trophoblast development, such as *GATA2* and *GATA3* (*Ralston et al., 2010*), and human-specific regulators, such as *VGLL1* (*Soncin et al., 2018*). Lastly, because recent studies have highlighted a transcriptional overlap between trophoblast and amnion cells (*Guo et al., 2021*; *Io et al., 2021*; *Zhao et al., 2021*), we examined whether genes reported to be expressed by amnion cells were upregulated in our dataset. We found that most of the amnion-

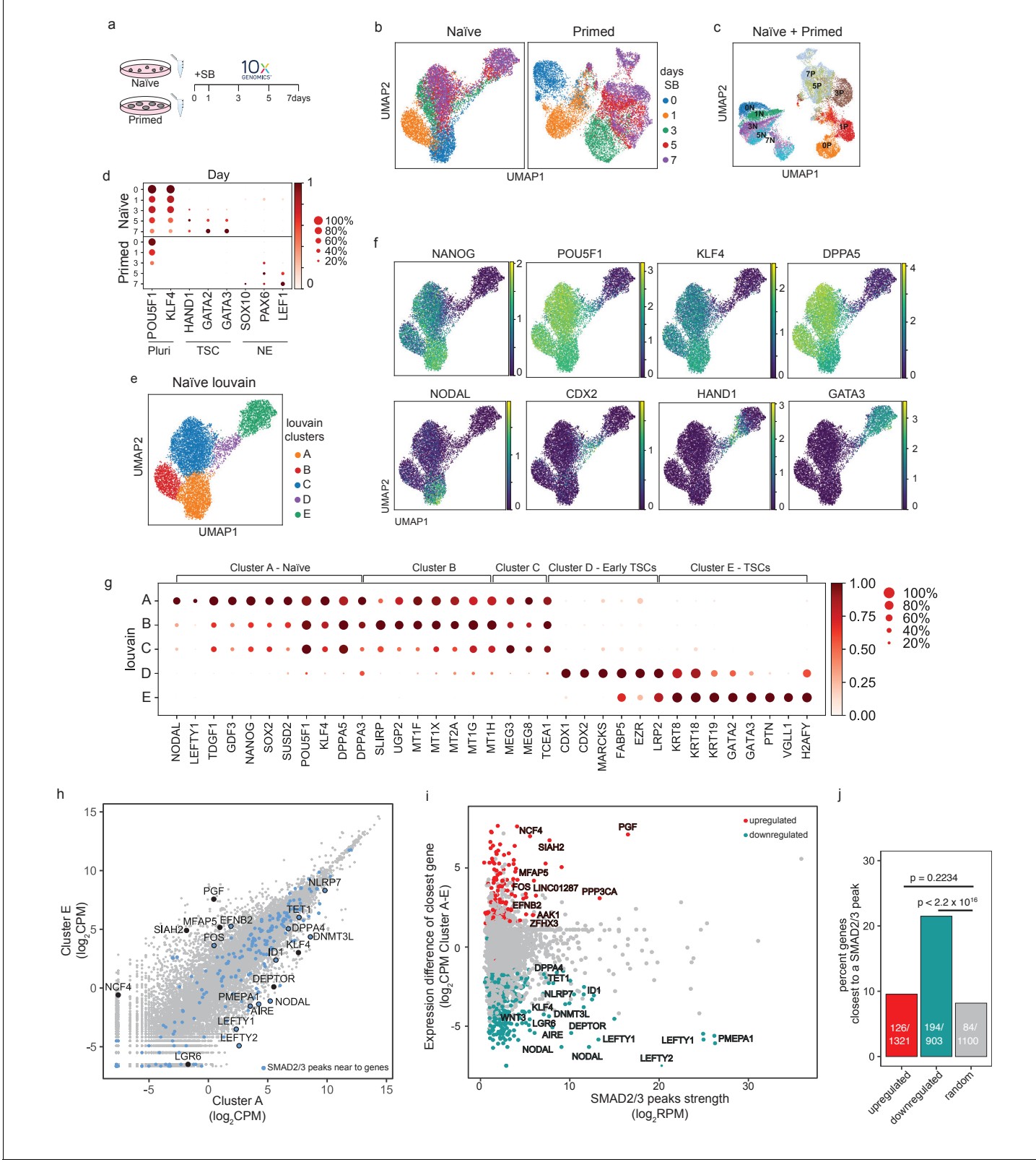

**Figure 4.** Single-cell transcriptional analysis reveals a trophoblast-like population arising in response to TGFβ inhibition in naïve hPSCs. (a) Overview of the experimental procedure. Naïve and primed hPSCs were cultured in the presence of SB-431542 (SB), a potent TGFβ inhibitor, and samples were collected at days 0, 1, 3, 5, and 7. Single-cell transcriptomes were obtained by 10X sequencing. (b) UMAP visualisation of naïve and primed cells during the SB time-course experiment, separated by days of treatment. (c) UMAP visualisation of the combined naïve and primed data set, separated by days

*Figure 4 continued on next page*

Figure 4 continued

of SB treatment (indicated by the number in the labels). N, naïve; P, primed. (**d**) Dot plot of selected gene expression values in naïve and primed cells during the SB time-course experiment, plotted by days of treatment (in rows). Each dot represents two values: mean expression within each category (visualised by colour) and fraction of cells expressing the gene (visualised by the size of the dot). Genes are indicative of pluripotent cells (Pluri), trophoblast stem cells (TSC), and neuroectoderm cells (NE). (**e**) UMAP visualisation of naïve hPSCs during the SB time-course experiment, separated by Louvain clustering (five clusters, A to E). (**f**) UMAP visualisation of naïve cells during the SB time-course experiment, showing the relative expression of pluripotency markers, *NANOG*, *POU5F1*, *KLF4*, and *DPPA5*; TGFβ effectors, *NODAL*; and trophoblast markers, *CDX2*, *HAND1*, *GATA3*. (**g**) Dot plot of expression values in naïve cells during the SB time-course experiment, separated by the five Louvain clusters. The genes shown represent a subset of the top 25 differentially expressed genes between the five clusters, as reported in *Figure 5—figure supplement 1e*. Each dot represents two values: mean expression within each category (visualised by colour) and fraction of cells expressing the gene (visualised by the size of the dot). (**h**) Scatter plot reporting pseudobulk RNA-seq values (from 10X data) for cells in Louvain clusters A and E. Each dot represents one gene. Genes that have SMAD2/3 ChIP-seq peaks ($\log_2$ RPM > 5) within 12 kb of their transcription start site (TSS) are highlighted in blue and annotated. Several differentially expressed genes that are the closest gene to a SMAD2/3 peak (but are further away than 12 kb) are also named. (**i**) Scatter plot showing SMAD2/3 ChIP-seq peak strength ($\log_2$ RPM) versus the expression difference (cluster A – cluster E; $\log_2$ CPM) of the gene nearest to the SMAD2/3 peak. Upregulated genes, red; downregulated genes, green. (**j**) SMAD2/3 peaks were annotated with their nearest genes. Bar plot showing the percentage of genes that are the closest gene to a SMAD2/3 peak for genes that are upregulated (red) or downregulated (green) between cells in clusters A and E. A randomly selected set of control genes are shown in grey. The number of closest genes and the set size are reported within the bars. Statistical testing was performed using Chi-square test with Yates continuity and Bonferroni multiple testing correction.

The online version of this article includes the following figure supplement(s) for figure 4:

**Figure supplement 1.** Single-cell transcriptional analysis reveals different trajectories between naïve and primed hPSCs following TGFβ inhibition.

associated genes examined were not detectable in any of the clusters (*Figure 4—figure supplement 1g*). Some markers, such as *CTSV* and *TPM1*, are expressed in both amnion and trophoblast, and as expected were upregulated in cluster E (*Figure 4—figure supplement 1g*). Although it is currently challenging to separate the transcriptional profiles of trophoblast and amnion cells, this analysis suggests that TGFβ inhibition of naïve hPSCs in these conditions does not promote the induction of reported amnion cell markers. Taken together, these results confirm that TGFβ inhibition downregulates a pluripotency program and enables trophectoderm differentiation from naïve hPSCs.

To dissect the impact of TGFβ pathway inhibition on the transcriptional changes, we overlapped cluster A and E gene expression profiles with SMAD2/3 ChIP-seq peaks. We found that a small subset of differentially expressed genes have a nearby SMAD2/3 peak (*Figure 4h*). Of note, many of the strongest peaks are close to differentially expressed genes, and this was especially clear for genes that are downregulated upon SB treatment (*Figure 4i*). Interestingly, among the downregulated genes, we found that SMAD2/3 bind within 12 kb of the transcriptional start sites of TGFβ downstream effectors (*NODAL*, *LEFTY1/2*, *PMEPA1*), key genes associated with naïve pluripotency (*DNMT3L*, *DPPA4*, *AIRE*, *ID1*), genes reported to inhibit trophoblast differentiation (*NLRP7*, *TET1*) (*Alici-Garipcan et al., 2020*; *Dawlaty et al., 2011*; *Koh et al., 2011*; *Mahadevan et al., 2014*), and also near to distal enhancers for other factors, such as *KLF4* and *DEPTOR* (*Figure 4h*). Although less prevalent, we also found SMAD2/3 binding sites close to some genes that are transcriptionally upregulated between cluster A and E, including *EFNB2* and *FOS*, and to enhancers close to *PGF* and *MFAP5*. To further assess the significance of this association, we tested how often differentially expressed genes between clusters A and E are the closest gene to a SMAD2/3 peak. Strikingly, 21% of downregulated genes are the closest gene to a SMAD2/3 binding site, which is significantly higher than the 7% of genes in a randomly-selected group of size-matched control genes ($p < 2.2 \times 10^{16}$, *Figure 4j*). These results suggest that the downregulation of pluripotency-associated genes following TGFβ inhibition is functionally linked to the loss of SMAD2/3 binding.

Taken together, scRNA-seq in primed and naïve cells shows that both developmental stages rely on TGFβ signalling to maintain their undifferentiated state but, upon pathway inhibition, each cell type diverges towards different trajectories. Primed cells differentiate into neuroectoderm cells whereas, in contrast, naïve cells exit pluripotency and acquire a TSC-like fate expressing trophoblast markers and this is triggered by the deregulation of target genes that are downstream of SMAD2/3.

# TGFβ inhibition in naïve hPSCs recapitulates the transcriptome of early trophoblast specification in human embryos

Having established that naïve hPSCs respond to TGFβ inhibition by shutting down the naïve pluripotency network, thereby allowing the onset of trophoblast differentiation, we next investigated whether this differentiation process follows a developmental trajectory. To do this, we applied diffusion pseudotime to our 10X scRNA-seq data (*Figure 5—figure supplement 1a*) and examined the pseudotime trajectory across the Louvain clusters (*Figure 5a*). Consistent with the prior UMAP analysis, we found that the time points (days) and the clusters progressively populate the trajectory following a similar pattern from cluster A, through B and C, towards a transition population in cluster D, and lastly the more differentiated counterpart in cluster E (*Figure 5a*). Overlaying the diffusion pseudotime maps with the expression of known markers reveals the initial downregulation of pluripotency genes, such as *NANOG*, was followed by a sequential upregulation of trophoblast markers, such as *CDX2*, *HAND1*, and *GATA3* (*Figure 5b*; *Figure 5—figure supplement 1b*). Interestingly, the transitional cell population in cluster D contains a substantial proportion of cells (~15–25%) that co-express low levels of the pluripotency gene *POU5F1* and trophoblast markers, such as *CDX2* and *HAND1* (*Figure 5c*). We confirmed this co-expression at the protein level using immunofluorescent microscopy (*Figure 5—figure supplement 1c*). These results indicate that trophoblast cells arise in the population through the transition of pluripotent cells to a trophoblast fate.

To further investigate the transition from naïve pluripotency to trophoblast specification, we compared our scRNA-seq data to human embryo transcriptional datasets (*Xiang et al., 2020*). Correlation analysis showed that cells in clusters A, B, and C are transcriptionally closest to epiblast cells, in keeping with their undifferentiated status (*Figure 5—figure supplement 1d*). The transitional population classified as cluster D has the highest correlation with ICM and TE (*Figure 5—figure supplement 1d*). Cells in cluster E have the highest correlation with trophoblast derivatives from the pre- and early-postimplantation embryo (*Figure 5—figure supplement 1d*).

We next focussed our analysis on the main pluripotent cell population (cluster A), the transitioning cells (cluster D) and the differentiated cells (cluster E). We compared these clusters with the embryo cell types that showed the highest transcriptional correlations to them (*Figure 5d*; *Figure 5—figure supplement 1d*). Visualising single cell transcriptomes for each cell type on a PCA plot revealed there was a good overlap between our stem cell differentiation series and the embryo lineages (*Figure 5e*), further supporting a transition from EPI to the trophoblast lineage. We then used the Wilcoxon Rank Sum test to identify marker genes for each embryo lineage and examined the expression pattern of those genes in cells across clusters A, D, and E. Interestingly, the two datasets have remarkably similar expression patterns, whereby the progression from clusters A to D to E closely resembles the transcriptional changes from EPI to trophoblast (*Figure 5f*). Among the top 20 genes per cluster (*Figure 5—figure supplement 1e*), we found genes, such as *NANOG* and *DPPA5* for cluster A / EPI, and trophoblast markers, such as *VGLL1* and *PGF* for cluster E / trophoblast, and confirmed their expression at the single cell level over the differentiation pseudotime (*Figure 5g*). Taken together, these results reveal that TGFβ inhibition of naïve hPSCs causes the cells to initiate a differentiation programme from pluripotency to TE-like cells and trophoblast derivatives, activating transcriptional identities similar to the embryo counterpart.

## Discussion

Here, we show that TGFβ/Activin/Nodal signalling is active in naïve hPSCs and that this pathway is required to maintain the cells in an undifferentiated state. These findings, therefore, establish that there is a continuum for TGFβ signalling function in pluripotency spanning a developmental window from naïve to primed states (*Figure 5h*).

Until now, the role of TGFβ signalling in naïve hPSCs has been unclear. Activators of this pathway are often included in naïve hPSCs culture formulations (*Bayerl et al., 2021*; *Chan et al., 2013*; *Theunissen et al., 2014*), suggesting that this pathway could be necessary to maintain pluripotency. Accordingly, we show here that naïve hPSCs transcribe high levels of endogenous TGFβ ligands and receptors, and the pathway is activated in standard naïve cell growth conditions as demonstrated by the phosphorylation status of SMAD2/3. These findings help to interpret previous observations from several studies. For example, when testing different culture formulations, the removal of Activin from 5i/L/A conditions led to an increase in the spontaneous differentiation of naïve hPSCs, and also

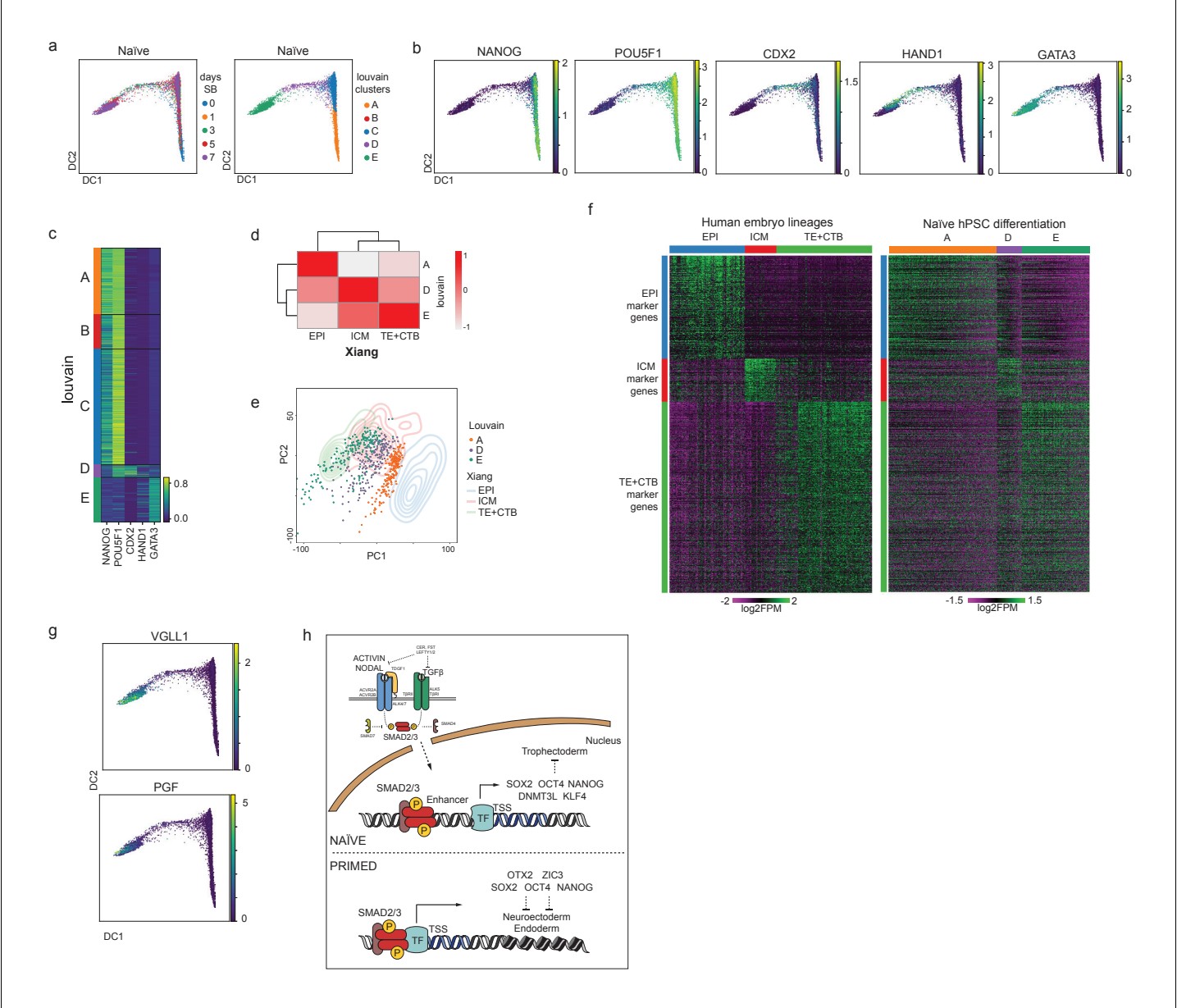

**Figure 5.** Differentiation of TGFβ-inhibited naïve hPSCs transcriptionally recapitulates early trophectoderm specification in human embryos. (a) Diffusion maps of naïve cells during the SB time-course experiment, separated by days of treatment (left) and Louvain clustering (right). (b) Overlay of the diffusion maps with the relative expression of pluripotency markers *NANOG*, and *POU5F1*, and trophoblast markers *CDX2*, *HAND1*, *GATA3*. (c) Heatmap of the expression values of genes reported in (b) separated by the Louvain clusters. Note the overlap in the expression of pluripotency and trophoblast markers in cells within cluster D. (d) Correlation plot between pseudobulk data from Louvain clusters A/D/E and EPI (Epiblast), ICM (Inner Cell Mass), and TE+CTB (Trophectoderm+Cytotrophoblast) from cultured human pre-gastrulation embryos (*Xiang et al., 2020*). (e) PCA plot overlapping 200 randomly selected cells from each of the Louvain clusters A/D/E (individual dots) and data from 3D-cultured human pre-gastrulation embryos (*Xiang et al., 2020*), based on EPI, ICM, and TE+CTB cells (contour lines). PC1 variance 2.15, PC2 variance 1.41. (f) Heatmaps visualising the expression of genes in EPI, ICM, and TE+CTB (*Xiang et al., 2020*) and cells in Louvain clusters A/D/E. Note that the genes are in the same order for both plots. (g) Diffusion maps of naïve cells during the SB time-course experiment showing the relative expression of CTB markers – *VGLL1* and *PGF*. (h) We propose there is a continuum of TGFβ/Activin/Nodal signalling that spans a developmental window of human pluripotent states from naïve to primed. In both states, active TGFβ signalling promotes the expression of common pluripotency genes, such as *NANOG* and *POU5F1*, and contributes to the maintenance of pluripotency. SMAD2/3 are additionally required in naïve hPSCs to sustain the expression of naïve pluripotency factors, including *KLF4* and *DNMT3L*. Inactivating TGFβ signalling in naïve hPSCs leads to the downregulation of pluripotency genes, thereby enabling the induction of trophoblast differentiation.

The online version of this article includes the following figure supplement(s) for figure 5:

*Figure 5 continued on next page*

*Figure 5 continued*

**Figure supplement 1.** Pseudotime trajectories of TGFβ-inhibited naïve hPSCs recapitulates early trophectoderm specification in human embryos.

to the reduced expression of naïve genes, including *NANOG* and *KLF4* (*Theunissen et al., 2014*). Furthermore, supplementing HENSM media with Activin caused naïve hPSCs to express higher levels of *KLF17, DNMT3L,* and *DPPA3* (genes that are confirmed as SMAD2/3 targets in our study) and elevated *POU5F1* distal enhancer activity, compared to the same conditions without Activin (*Bayerl et al., 2021*). In addition to the effect on established naïve cell lines, Activin also enhanced the kinetics of primed to naïve hPSCs reprogramming (*Theunissen et al., 2014*). At the time, the authors speculated that Activin prolongs primed hPSCs in a state that is amenable to naïve reprogramming. Based on the results from our study, we propose that TGFβ signalling is required to maintain pluripotency in cells throughout primed to naïve cell reprogramming and additionally enforce the expression of genes that promote naïve hPSCs. Thus, TGFβ/Activin/Nodal signalling helps to stabilise naïve pluripotency and the addition of Activin to naïve induction and maintenance conditions is predicted to be beneficial.

At the molecular level, our analysis showed that SMAD2/3, the DNA-binding effectors downstream of TGFβ/Activin/Nodal signalling, occupied genomic sites that were common to both naïve and primed hPSCs, in addition to a large set of cell type-specific sites. Shared target genes included core pluripotency factors, such as *NANOG*, in addition to factors that are canonical targets, such as *LEFTY1/2* and *SMAD7*. Disrupting TGFβ signalling in naïve and primed hPSCs caused the rapid downregulation of these common target genes, indicating the presence of shared gene regulatory networks between the two pluripotent states. We additionally identified a large set of genes that were targeted by SMAD2/3 in naïve hPSCs but not in primed hPSCs. This set of genes included *KLF4, TFAP2C,* and *DNMT3L*, which are important regulators of naïve pluripotency (*Bayerl et al., 2021*; *Pastor et al., 2018*), and we demonstrated that their expression levels were also sensitive to TGFβ pathway inhibition. These findings indicate that TGFβ/Activin/Nodal signalling functions in naïve hPSCs to reinforce the expression of key genes that promote naïve pluripotency, rather than to repress differentiation-promoting factors. Previous studies suggest that TGFβ/Activin/Nodal signalling may regulate NANOG expression in human embryos (*Blakeley et al., 2017*). It will be important to determine in the future whether the signalling requirements we uncover in naïve hPSCs could also be operating in pluripotent cells of human embryos. If so, then existing naïve hPSCs may serve as a useful cell model in which to investigate the mechanisms of signalling pathways that are relevant for early human development, alternatively, if this shows distinctions it may point to ways in which current in vitro conditions may need to be further refined to more closely recapitulate the pre-implantation embryonic epiblast in the embryo. Importantly, genetic studies in the mouse have established a key function for Nodal-SMAD2/3 signalling in maintaining the pluripotent state of post-implantation epiblast and in the formation of the primitive streak during gastrulation (*Brennan et al., 2001*; *Varlet et al., 1997*). Concerning pre-implantation stages, TGFβ/Activin/Nodal signalling appears to play a role in the regionalisation of the extraembryonic endoderm. However, a function in the early epiblast remains elusive, thereby suggesting the existence of species divergence regarding TGFβ/Activin/Nodal signalling function during early development.

Our experiments also uncovered a widespread relocalisation in the genomic sites that are occupied by SMAD2/3. By integrating our datasets with chromatin and transcription factor profiles, we found that SMAD2/3 binding was enriched at active enhancers in naïve cells, yet predominantly at promoters in primed cells. This redistribution mirrors changes in OCT4, SOX2 and NANOG occupancy, whereby sites bound by SMAD2/3 only in naïve hPSCs are also preferentially occupied by OSN in naïve compared to primed cells. These findings predict that SMAD2/3 and OSN integrate signalling and transcription factor inputs in naïve pluripotency, similar to the functional interaction between SMAD2/3 and NANOG in primed hPSCs (*Brown et al., 2011*; *Xu et al., 2008*). Together, these results establish that TGFβ signalling is a core feature that is closely integrated within the transcriptional network of naïve hPSCs.

Finally, our single-cell analysis revealed that naïve and primed hPSCs depart along different trajectories following TGFβ inhibition. Primed hPSCs differentiated rapidly into neuroectoderm following TGFβ inhibition, which is consistent with previous studies (*Smith et al., 2008*). In contrast, naïve

hPSCs upregulated trophoblast-associated genes after several days of TGFβ inhibition. The divergent routes taken by naïve and primed hPSCs could be due to their different developmental states and differentiation potential. In keeping with their preimplantation epiblast identity, naïve hPSCs can differentiate efficiently into trophoblast and hypoblast, and are required to transition through a process of capacitation to acquire the competency to respond directly to signals that promote postimplantation germ layer induction. In contrast, primed hPSCs are more similar to early postimplantation epiblast, and therefore less efficiently make trophoblast and hypoblast. Additionally, the presence of different factors and inhibitors in the naïve and primed hPSC culture media could also affect their responses to TGFβ inhibition. Importantly, not all of the cells in the inhibitor-treated naïve cultures differentiated uniformly over the first few days. Thus, we speculate that the presence of small molecules inhibiting MEK, GSK3β, and PKC with the addition of LIF can attenuate the effect of TGFβ inhibition (*Guo et al., 2021*). Notably, a TGFβ inhibitor is a common component of human TSC medium (*Okae et al., 2018*), which suggests that TGFβ signalling may act to limit trophoblast self-renewal or proliferation. TGFβ inhibitors are also a component in the conditions that can convert naïve hPSCs to TSCs (*Castel et al., 2020*; *Cinkornpumin et al., 2020*; *Dong et al., 2020*; *Liu et al., 2020*). Here, the inhibitor might be functioning in two ways: to induce the exit from naïve pluripotency, and to promote trophoblast cell growth. Unexpectedly, our single cell analysis revealed that following TGFβ inhibition, naïve cells acquire a transcriptional identity closest to ICM and early TE, marked, for example, by transient *CDX2* expression, and then the cells undergo further differentiation into trophoblast cell types. *CDX1* and *CDX2* are expressed transiently in primate trophoblast development including the pre-implantation TE in human blastocysts (*Niakan and Eggan, 2013*), but are not expressed in embryo-derived TSCs or naïve hPSC-derived TSCs, which are more similar to post-implantation trophoblast (*Castel et al., 2020*; *Dong et al., 2020*; *Okae et al., 2018*). By capturing pre-implantation TE-like cells, our scRNA-seq data could, therefore, shed light on the transcriptional changes that occur during the early stages of human trophoblast specification.

To conclude, our results establish a central role for TGFβ/Activin/Nodal signalling in protecting human pluripotent stem cells against differentiation. This knowledge will be useful to establish culture conditions allowing the derivation and production in vitro of different cell types constituting the human embryo. In addition, modulation of TGFβ could play a key role in the early human embryo and could be useful for improving culture conditions used to grow human embryos in vitro.

## Materials and methods

**Key resources table**

| Reagent type (species) or resource | Designation | Source or reference | Identifiers | Additional information |
|---|---|---|---|---|
| Gene (*Homo sapiens*) | *SMAD2* | GenBank | Gene ID:4087 | |
| Gene (*Homo sapiens*) | *SMAD3* | GenBank | Gene ID:4088 | |
| Gene (*Homo sapiens*) | *NODAL* | GenBank | Gene ID:4838 | |
| Gene (*Homo sapiens*) | *POU5F1* | GenBank | Gene ID:5460 | |
| Gene (*Homo sapiens*) | *NANOG* | GenBank | Gene ID:79923 | |
| Gene (*Homo sapiens*) | *KLF4* | GenBank | Gene ID:9314 | |
| Gene (*Homo sapiens*) | *GATA3* | GenBank | Gene ID:2625 | |
| Gene (*Homo sapiens*) | *CDX2* | GenBank | Gene ID:1045 | |
| Cell line (*Homo sapiens*) | WA09/H9 | WiCell | RRID:CVCL_9773 | Human Embryonic Stem Cell line |

*Continued on next page*

*Continued*

| Reagent type (species) or resource | Designation | Source or reference | Identifiers | Additional information |
|---|---|---|---|---|
| Cell line (*Homo sapiens*) | HNES1 | DOI:10.1016/j.stemcr.2016.02.005 | RRID:CVCL_9R98 | Human Embryonic Stem Cell line |
| Transfected construct (*Homo sapiens*) | SMAD2 shRNA | Sigma | TRCN0000010477 | CCGGCAAGTACTCCTT GCTGGATTGCTCGAGCAA TCCAGCAAGGAGTA CTTGTTTTTG |
| Transfected construct (*Homo sapiens*) | SMAD3 shRNA | Sigma | TRCN0000330055 | CCGGGCCTCAGTGACA GCGCTATTTCTCGAG AAATAGCGCTGT CA CTGAGGCTTTTTG |
| Antibody | SMAD2/3 (Goat polyclonal) | R&D | AF3797; RRID:AB_2270778 | ChIP 10 µg |
| Antibody | SMAD2/3 (Rabbit monoclonal Biotinylated) | Cell signaling | 12470S; RRID:AB_2797930 | WB 1:5,000 |
| Antibody | pSMAD2 (Rabbit monoclonal) | Cell signaling | 3108S; RRID:AB_490941 | WB 1:1,000 |
| Antibody | α-Tubulin (Mouse monoclonal) | Sigma | T6199; RRID:AB_477583 | WB 1:10,000 |
| Antibody | Vinculin (Mouse monoclonal) | Sigma | SAB4200080; RRID:AB_10604160 | WB 1:20,000 |
| Antibody | GATA3 (D13C9) (Rabbit monoclonal) | Cell signaling | 5852S; RRID:AB_10835690 | IF 1:100 |
| Antibody | NANOG (Rabbit polyclonal) | Abcam | ab21624; RRID:AB_446437 | IF 1:100 |
| Antibody | OCT3/4 (C-10) (Mouse monoclonal) | Santa Cruz Biotechnology | sc-5279; RRID:AB_628051 | IF 1:200 |
| Antibody | HAND1 (Goat polyclonal) | R&D | AF3168; RRID:AB_2115853 | IF 1:200 |
| Antibody | CK7 (Rabbit monoclonal) | Abcam | ab68459; RRID:AB_1139824 | IF 1:100 |
| Antibody | CK19 (Mouse monoclonal) | Abcam | ab7754; RRID:AB_306048 | IF 1:100 |
| Antibody | SDC1 (Mouse monoclonal) | Abcam | ab34164; RRID:AB_778207 | IF 1:100 |
| Antibody | HLA-G (Mouse monoclonal) | Santa Cruz Biotechnology | sc-21799; RRID:AB_627938 | IF 1:100 |
| Antibody | Secondary antibodies | | | *Supplementary file 1* |
| Sequence-based reagent | RT-qPCR primers | | | *Supplementary file 2* |
| Chemical compound, drug | SB-431542 | Tocris | 1614 | 10–20 µM |
| Software, algorithm | Fiji | https://doi.org/10.1038/nmeth.2019 | RRID:SCR_002285 | |
| Software, algorithm | GraphPad Prism | http://www.graphpad.com/ | RRID:SCR_002798 | |
| Software, algorithm | SeqMonk | http://www.bioinformatics.babraham.ac.uk/projects/seqmonk/ | RRID:SCR_001913 | |
| Software, algorithm | RStudio | http://www.rstudio.com/ | RRID:SCR_000432 | R packages used specified in Materials and Methods |

*Continued on next page*

*Continued*

| Reagent type (species) or resource | Designation | Source or reference | Identifiers | Additional information |
|---|---|---|---|---|
| Software, algorithm | CellRanger | https://support.10xgenomics.com/single-cell-gene-expression/software/pipelines/latest/what-is-cell-ranger | RRID:SCR_017344 | |
| Software, algorithm | Scanpy | https://github.com/theislab/scanpy | RRID:SCR_018139 | |
| Other | DAPI stain | Invitrogen | D1306 | 0.1 µg/ml |

## Cell culture

Transgene-reset WA09/H9 NK2, embryo-derived HNES1 and chemically reset cR-H9 naïve hPSCs (*Guo et al., 2017*; *Guo et al., 2016*; *Takashima et al., 2014*) were kindly provided by Dr. Austin Smith with permission from WiCell and the UK Stem Cell Bank Steering Committee. Cells were maintained in t2iLGö (*Takashima et al., 2014*) or in PXGL (*Bredenkamp et al., 2019b*; *Rostovskaya et al., 2019*) in hypoxia (5% O$_2$) at 37°C. The N2B27 base medium contained a 1:1 mixture of DMEM/F12 and Neurobasal, 0.5X N-2 supplement, 0.5X B-27 supplement, 2 mM L-Glutamine, 0.1 mM β-mercaptoethanol (all from ThermoFisher Scientific), 0.5X Penicillin/Streptomycin. For t2iLGö, the base medium was supplemented with 1 µM PD0325901, 1 µM CHIR99021, 20 ng/ml human LIF (all from WT-MRC Cambridge Stem Cell Institute) and 2 µM Gö6983 (Tocris). For PXGL, N2B27 medium was supplemented with 1 µM PD0325901, 2 µM XAV939, 2 µM Gö6983 and 10 ng/ml human LIF. Naïve hPSCs were maintained on a layer of irradiated mouse fibroblasts that were seeded at a density of two million cells per six-well plate. All experiments have been performed on naïve hPSCs that were grown in the absence of mouse fibroblasts for at least two passages using Growth Factor Reduced Matrigel-coated plates (Corning). For TGFβ inhibition experiments, 20 µM SB-431542 (Tocris) was added to the medium for the specified length of the experiment.

Conventional (primed) WA09/H9 (*Thomson et al., 1998*) were maintained in E8 medium as previously described (*Chen et al., 2011*) in DMEM/F12, 0.05% Sodium Bicarbonate, 2X Insulin-Transferrin-Selenium solution (all from ThermoFisher Scientific), 64 µg/ml L-ascorbic acid 2-phosphate (LAA) (Sigma), 1X Penicillin/Streptomycin (WT-MRC Cambridge Stem Cell Institute), 25 ng/ml FGF2 (Hyvönen Group, Dept of Biochemistry), and 2 ng/ml TGFβ (BioTechne) on 10 µg/ml of Vitronectin XF-coated plates (StemCell Technologies) at 37°C. For TGFβ inhibition experiments, 10 µM SB-431542 (Tocris) was added to the media for the specified length of the experiment.

TSCs, EVT, and STB cells were generated as previously described (*Dong et al., 2020*) with some modifications as follows. WA09/H9 NK2 naïve hPSCs were treated with 10 µM SB-431542 (Tocris) in t2iLGö media for 5 days. Cells were dissociated with TrypLE (ThermoFisher Scientific) and single cells were seeded on Collagen IV-coated plates (5 µg/ml; Sigma) in TSC media (*Okae et al., 2018*) comprising of DMEM/F12 supplemented with 0.1 mM β-mercaptoethanol, 0.2% FBS, 0.5% Penicillin/Streptomycin, 0.3% BSA, 1% ITS-X (all from ThermoFisher Scientific), 1.5 µg/ml L-ascorbic acid (Sigma), 50 ng/ml EGF (Peprotech), 2 µM CHIR99021 (WT-MRC Cambridge Stem Cell Institute), 0.5 µM A83-01 (Tocris), 1 µM SB-431542 (Tocris), 0.8 mM VPA (Sigma), and 5 µM Y-27632 (Cell Guidance Systems) in 5% CO$_2$. The media was changed every 2 days and cells were passaged with TrypLE when ~80% confluent. To induce EVT differentiation, dissociated naïve-derived TSCs were seeded onto plates pre-coated with 1 µg/ml of Collagen IV (Sigma) in EVT basal media comprising DMEM/F12 with 0.1 mM β-mercaptoethanol, 0.5% Penicillin/Streptomycin, 0.3% BSA, 1% ITS-X (all ThermoFisher Scientific), 7.5 µM A83-01 (Tocris), 2.5 µM Y27632 (Cell Guidance Systems) and supplemented with 4% KSR (ThermoFisher Scientific) and 100 ng/ml NRG1 (Cell Signalling). Matrigel (Corning) was added at 2% final concentration shortly after resuspending the cells in the media. On day 3, the media was replaced with EVT basal media supplemented with 4% KSR (ThermoFisher Scientific), and Matrigel (Corning) was added at 0.5% final concentration. On day 6, the media were replaced with EVT basal medium, plus 0.5% Matrigel (Corning). EVTs were cultured for two more days and then collected for analysis. To induce STB differentiation, dissociated TSCs were seeded in

STB media comprising DMEM/F12 supplemented with 0.1 mM β-mercaptoethanol, 0.5% Penicillin/ Streptomycin, 0.3% BSA, 1% ITS-X (all ThermoFisher Scientific), 2.5 µM Y-27632 (Cell Guidance Systems), 50 ng/ml EGF (Peprotech), 2 µM Forskolin (R&D) and 4% KSR (ThermFisher Scientific) in ultra-low attachment plates to form cell aggregates in suspension. Fresh media was added on day 3, and samples were collected for analysis on day 6.

Authentication of hPSCs was achieved by confirming the expression of pluripotency genes and protein markers (NANOG and OCT4). Cells were routinely verified as mycoplasma-free using broth and PCR-based assays. The cell lines are not on the list of commonly misidentified cell lines (International Cell Line Authentication Committee).

## Western blotting

For whole cell lysates, cells were washed once in D-PBS and resuspended in ice cold RIPA buffer (150 mM NaCl, 50 mM Tris, pH8.0, 1% NP-40, 0.5% sodium deoxycholate, 0.1% sodium dodecyl sulfate) containing protease and phosphatase inhibitors for 10 min. Protein concentration was quantified by a BCA assay (Pierce) following the manufacturer's instructions using a standard curve generated from BSA and read at 600 nm on an EnVision 2104 plate reader. Samples were prepared by adding 4x NuPAGE LDS sample buffer (ThermoFisher Scientific) plus 1% β-mercaptoethanol and heated at 95°C for 5 min. 5–10 µg of protein per sample was run on a 4–12% NuPAGE Bis-Tris Gel (ThermoFisher Scientific) and then transferred to PVDF membrane by liquid transfer using NuPAGE Transfer buffer (ThermoFisher Scientific). Membranes were blocked for 1 hr at RT in PBS 0.05% Tween-20 (PBST) supplemented with 4% non-fat dried milk and incubated overnight at 4°C with primary antibodies diluted in the same blocking buffer, or 5% BSA in case of phosphor-proteins. After three washes in PBST, membranes were incubated for 1 hr at RT with horseradish peroxidase (HRP)-conjugated secondary antibodies diluted in blocking buffer, then washed a further three times before being incubated with Pierce ECL2 Western Blotting Substrate (ThermoFisher Scientific) and exposed to X-Ray Film. Membranes were probed with antibodies in *Supplementary file 1*. Relative quantification was performed using Fiji (ImageJ). Western blots were performed in three different lines, with the NK2 line in biological duplicate (*Figure 1*; *Figure 1—figure supplement 1* and *Figure 3—figure supplement 1*).

## RNA extraction and quantitative reverse transcription PCR (RT-qPCR)

Total RNA was extracted with the GeneElute Total RNA kit (Sigma). The on-column DNase digestion step was performed (Sigma) to remove any genomic DNA contamination. Of total RNA, 500 ng was used to synthesize cDNA with SuperScript II (ThermoFisher Scientific) using Random primers (Promega) following manufacturer's instructions. cDNA was diluted 30-fold and 2.5 µl was used to perform Quantitative PCR using Kapa SYBR fast Low-Rox (Sigma) in a final reaction volume of 7.5 µl on a QuantStudio 5 384 PCR machine (ThermoFisher Scientific). Samples were run in technical duplicate as two wells in the same qPCR plate and results were analysed using *PBGD/RPLP0* as housekeeping genes. All experiments were run in biological triplicate unless specified in the figure legends. Biological replicates were defined as separate experiments using the same line from three different passages performed at different times. All primer pairs were validated to ensure only one product was amplified and with a PCR efficiency of 100% (±10%). Primer sequences used are displayed in *Supplementary file 2*.

## SMAD2/3 iKD line and reprogramming

Validated short hairpin RNA (shRNA) against SMAD2 and SMAD3 were obtained from Sigma and the sequences are shown in the Key Resources Table. Construction and transfection of the sOPTiKD plasmid as well as cloning were carried out as described in *Bertero et al., 2018*. GeneJuice Transfection Reagent (Sigma) was used for transfection.

Primed SMAD2/3 inducible knockdown hPSCs were reprogrammed to a naïve state in 5i/L/A conditions (*Theunissen et al., 2014*). Primed hPSCs were dissociated into single cells with Accutase and 1.2 million cells per 10 cm tissue culture dish were plated in primed hPSCs media with 10 µM Y-27632 (Cell Guidance Systems) onto MEF seeded at a density of 4 million cells per 10 cm dish. The following day, media was changed to 5i/L/A comprising of a 1:1 mixture of DMEM/F12 and Neurobasal, 1X N-2 supplement, 1X B-27 supplement, 1% nonessential amino acids, 2 mM GlutaMAX, 50

U/ml and 50 µg/ml penicillin-streptomycin (all from ThermoFisher Scientific), 0.1 mM β-mercaptoe-thanol (Millipore), 50 µg/ml bovine serum albumin (ThermoFisher Scientific), 0.5% Knockout Serum Replacement (ThermoFisher Scientific), 20 ng/ml recombinant human LIF, 20 ng/ml ActivinA, 1 µM PD0325901 (all from WT-MRC Cambridge Stem Cell Institute), 1 µM IM-12, 1 µM WH-4–023, 0.5 µM SB590885 and 10 µM Y-27632 (all from Cell Guidance Systems). Cells were passaged with Accutase on days 5 and 10. Knockdown was induced by adding 1 µg/ml Tetracycline (Sigma) dissolved in Embryo Transfer Water (Sigma) to the media.

## Immunofluorescence

Cells were grown on glass coverslips coated with either Matrigel or Vitronectin XF and fixed with 4% PFA for 10 min at RT, rinsed twice with PBS, and permeabilised for 20 min at RT using PBS/0.25% Triton X-100 (Sigma). Cells were blocked for 30 min at RT with blocking solution (PBS-0.25% Triton X-100 plus BSA 1%). Primary and secondary antibodies (listed in *Supplementary file 1*) were diluted in blocking solution and incubated for 1 hr at 37°C. Cells were washed twice with blocking solution after each antibody staining, and stained with DAPI for 5 min at RT (0.1 µg/ml DAPI in PBS-0.1% Tri-ton). Finally, coverslips were mounted on slides using ProLong Gold antifade reagent (ThermoFisher Scientific) and imaged using an LSM 700 confocal microscope (Zeiss). To image STB cells, cell aggregates were collected by gentle centrifugation (100 x g for 30 s) and fixed in 4% PFA for 20 min. Cells were rinsed twice with PBS and resuspended in 100 µl of PBS and dried overnight on plus-charged slides (SuperFrost Plus Adhesion slides, Fisher Scientific). The area containing the dried cells was circled with a PAP pen and the cells were permeabilised for 5 min at RT with 100 µl of 0.1% Triton-X100 in PBS, and then blocked for 1 hr at RT with 100 µl of 0.1% Triton-X100 plus 0.5% BSA. Primary antibodies (listed in *Supplementary file 1*) were diluted in blocking solution and incubated overnight at 4°C. Cells were washed three times with blocking solution for 5 min, and stained with secondary antibodies for 1 hr at RT. Cells were then washed for 15 min with PBS, followed by a second PBS wash supplemented with DAPI (0.1 µg/ml) and a third with PBS. Finally, coverslips were mounted using ProLong Gold antifade reagent (ThermoFisher Scientific) and imaged using an LSM 700 confocal microscope (Zeiss). Images processed using the software Fiji (ImageJ). At least four different fields from each experiment were imaged and representative ones are shown in the figures.

## Chromatin immunoprecipitation (ChIP) sequencing

Chromatin immunoprecipitation (ChIP) was performed as previously described (*Brown et al., 2011*), using HEPES buffer containing 1% formaldehyde at room temperature, 10 mM Dimethyl 3,3′-dithio-propionimidate dihydrochloride (DTBP, Sigma) and 2.5 mM 3,3′-Dithiodipropionic acid di(N-hydro-ysuccinimide ester) (DSP, Sigma) for the crosslinking step. Experiments were performed on biological duplicates, carried out at different times with cells from two different passages. 10 µg of SMAD2/3 antibody (*Supplementary file 1*) was used per ChIP, and samples were purified using the iPure v2 bead kit (Diagenode). Libraries were constructed using the MicroPlex Library Preparation Kit v2 (Diagenode) following the manufacturer's instructions, 10 ng of input and all of the ChIP DNA was used as the starting material. Libraries were quantified using KAPA Library Quantification Kit (Roche) following the manufacturer's instructions and by BioAnalyser. Sequencing was performed at the Babraham Institute's Next-Generation Sequencing Facility. Equimolar amounts of each library were pooled, and eight samples were multiplexed on one lane of a NextSeq500 HighOutput 75 bp Single End run.

## Data processing

Reads were quality and adapter trimmed using Trim Galore! (version 0.5.0_dev, Cutadapt version 1.15), and aligned to GRCh38 using Bowtie 2 (version 2.3.2).

## Data analysis

All analyses were performed using SeqMonk (https://www.bioinformatics.babraham.ac.uk/projects/seqmonk/, version 1.46.0) or R (https://www.R-project.org/, version 4.0.2). For quantitation, read lengths were extended to 300 bp and regions of coverage outliers were excluded. SMAD2/3 peaks were called using a SeqMonk implementation of MACS (*Zhang et al., 2008*) with parameters p<10E-6, sonicated fragment size = 300. Peaks were called individually for both replicates and the

overlap of peaks used for annotation. Control regions were randomly selected from 700 bp tiles not overlapping excluded regions.

Differential binding analysis was performed using the R package Diffbind, and analysis of motifs that are relatively enriched in naïve compared to primed was performed using the MEME suit tool AME.

## Single-cell RNA-seq (10X chromium single cell)

H9 NK2 naïve and H9 primed hESCs were grown in presence of 20 µM (naïve) and 10 µM (primed) of SB-431542 for 7 days. Cells during the time-course were collected at day 0 (control, no SB) and at days 1, 3, 5, and 7 of treatment and dissociated with Accutase (ThermoFisher Scientific) for 5 min at 37°C in hypoxia (5% $O_2$), and resuspended until single-cell suspension was obtained. Accutase was blocked by adding PBS/BSA 0.5% with the respective media, and after a wash pellets were resuspended at a concentration of ~1000 cells/µl in the respective media. A total of 3000 cells/sample were loaded on a Chromium Chip B Single Cell following the manufacturer's instruction to generate Gel Beads-in-emulsion (GEMs) using a Reagent kit v3. Final Chromium Single Cell 3′ Gene Expression library was generated using standard Illumina paired-end constructs with P5 and P7 primers.

## Data analysis

Cell Ranger pipeline (version 3.0.2) was used to align reads to GRCh38 assembly and generate feature-barcode matrices for further gene expression analyses. Quality control, normalisation, dimensionality reduction analyses and all downstream analyses were carried out using the python-based library Scanpy (*Wolf et al., 2018*). Genes with read counts > 0 in at least three cells and cells expressing at least 200 genes were maintained for downstream analysis. Low-quality cells were removed based on the percentage of unique molecular identifiers (UMIs) mapping to the mitochondrial genome and the number of genes detected. Logarithmic normalisation was performed, highly variable genes were selected, the total number of UMIs per cell was regressed out from log-normalised data and the regressed expression values were scaled. The dimensionality reduction was performed using Principal Component Analysis (PCA) and the neighborhood graph of cells was calculated using the PCA representation of the scaled data matrix. Clustering was performed on scaled data using the Louvain method. This graph was embedded in two dimensions using Uniform Manifold Approximation and Projection (UMAP).

Transcriptional similarity was also quantified at origin and region resolution by estimating the connectivity of data manifold partitions within the partition-based graph abstraction (PAGA) framework. Cluster markers and differentially expressed genes were identified by applying the Wilcoxon-Rank-Sum test. In order to visualise the gradual variation in the transcriptional profile following the differentiation induced by the SB treatment, cells were represented as a pseudo-spatial dimension using the diffusion pseudotime method.

FPKM values for the ICM, EPI, and TE/CTB single-cell RNA-Seq datasets from *Xiang et al., 2020* (GSE136447) were extracted and log2 transformed. Similarly, log2 reads per 10 k values from the first 200 cells from the A, D, and E Louvain clusters were also prepared. For PCA analysis, the data was filtered to retain only genes which were expressed in more than 10% of cells in both datasets. PCA was used to separate the cells in the filtered Xiang et al. data, retaining the first and second principal components. The rotations from this analysis were then applied to the Louvain cluster data to project it into the same space (*Figure 5e*).

For overall correlation (*Figure 5d* and *Figure 5—figure supplement 1d*) the mean log2 FPKM for each condition from the Xiang et al. data was correlated with the summed, log2 FPM values from the Louvain clusters using Pearson's correlation. Only genes with log2 FPKM > 0.2 in any Xiang et al. dataset and raw counts > 2 in any Louvain cluster were used for the calculation.

For the single-cell heatmaps (*Figure 5f*), a Wilcoxon Rank Sum test was used to identify marker genes which were significantly (fdr < 0.05, Benjamini-Hochberg correction) enriched in one Xiang et al. scRNA-seq condition relative to the others. We then plotted a heatmap of the expression patterns of the marker genes (columns) in each cell (rows) for both the Xiang et al. and Louvain cluster data, with the cells being ordered by the group to which they belonged. Measures were per-gene z-score normalised log2 FPM.

## Acknowledgements

We thank Kristina Tabbada and Clare Murnane of the Babraham Institute Next Generation Sequencing Facility, and Felix Krueger from Babraham Bioinformatics for sequencing QC and mapping. We also thank Steven Leonard of the Wellcome Sanger Institute for pre-processing single cell RNA-seq data. We are very grateful to Vicente Perez-Garcia (Babraham Institute and the Centre for Trophoblast Research) for providing advice and reagents for characterising the naïve-derived human trophoblast cells. The work was supported by grants to PJR-G from the BBSRC (BBS/E/B/000C0421, BBS/E/B/000C0422) and the MRC (MR/T011769/1). AJC was supported by an MRC DTG Studentship (MR/J003808/1). The work was also supported by the European Research Council Grant New-Chol to LV and AO, the Cambridge Hospitals National Institute for Health Research Biomedical Research Center to LV and SB, the EU H2020 INTENS grant to DO, a Gates Cambridge PhD studentship to BTW, a JSPS Overseas Research Fellowship (201860446) and a Grant-in-Aid (16J08005) to SN, and core support grant from the Wellcome and Medical Research Council to the Wellcome – Medical Research Council Cambridge Stem Cell Institute. The work in KN's laboratory was supported by the Francis Crick Institute which receives its core funding from Cancer Research UK (FC001120), the Medical Research Council (FC001120), and the Wellcome Trust (FC001120). For the purpose of Open Access, the authors have applied a CC BY public copyright licence to any Author Accepted Manuscript version arising from this submission.

## Additional information

### Funding

| Funder | Grant reference number | Author |
|---|---|---|
| Biotechnology and Biological Sciences Research Council | BBS/E/B/000C0421 | Peter J Rugg-Gunn |
| Biotechnology and Biological Sciences Research Council | BBS/E/B/000C0422 | Peter J Rugg-Gunn |
| Medical Research Council | MR/T011769/1 | Peter J Rugg-Gunn |
| Medical Research Council | MR/J003808/1 | Amanda J Collier |
| European Research Council | New-Chol | Ludovic Vallier Anna Osnato |
| European Research Council | INTENS | Daniel Ortmann |
| Gates Cambridge Trust | | Brandon T Wesley |
| Japan Society for the Promotion of Science | 201860446 | Shota Nakanoh |
| Japan Society for the Promotion of Science | 16J08005 | Shota Nakanoh |
| Cancer Research UK | FC001120 | Kathy K Niakan |
| Medical Research Council | FC001120 | Kathy Niakan |
| Wellcome Trust | FC001120 | Kathy Niakan |
| Cambridge Hospitals National Institute for Health Research Biomedical Research Center | | Ludovic Vallier Stephanie Brown |

The funders had no role in study design, data collection and interpretation, or the decision to submit the work for publication.

### Author contributions

Anna Osnato, Conceptualization, Data curation, Formal analysis, Validation, Investigation, Visualization, Writing - original draft, Writing - review and editing; Stephanie Brown, Conceptualization, Validation, Investigation, Visualization, Writing - review and editing; Christel Krueger, Simon Andrews, Formal analysis, Visualization, Writing - review and editing; Amanda J Collier, Shota Nakanoh,

Mariana Quiroga Londoño, Brandon T Wesley, A Sophie Brumm, Investigation, Writing - review and editing; Daniele Muraro, Data curation, Formal analysis, Writing - review and editing; Kathy K Niakan, Supervision, Funding acquisition, Writing - review and editing; Ludovic Vallier, Conceptualization, Supervision, Funding acquisition, Writing - original draft, Project administration, Writing - review and editing; Daniel Ortmann, Conceptualization, Supervision, Validation, Investigation, Writing - original draft, Project administration, Writing - review and editing; Peter J Rugg-Gunn, Conceptualization, Supervision, Funding acquisition, Investigation, Writing - original draft, Project administration, Writing - review and editing

## Author ORCIDs

Anna Osnato (iD) https://orcid.org/0000-0001-5241-1512
Christel Krueger (iD) https://orcid.org/0000-0001-5601-598X
Simon Andrews (iD) https://orcid.org/0000-0002-5006-3507
Amanda J Collier (iD) https://orcid.org/0000-0003-1137-6874
Mariana Quiroga Londoño (iD) https://orcid.org/0000-0003-2352-0773
Kathy K Niakan (iD) https://orcid.org/0000-0003-1646-4734
Ludovic Vallier (iD) https://orcid.org/0000-0002-3848-2602
Peter J Rugg-Gunn (iD) https://orcid.org/0000-0002-9601-5949

## Decision letter and Author response

Decision letter https://doi.org/10.7554/eLife.67259.sa1
Author response https://doi.org/10.7554/eLife.67259.sa2

## Additional files

### Supplementary files

• Supplementary file 1. Table of antibodies used for Western Blots, Immunofluorescence and Chromatin Immunoprecipitation and relative dilutions.

• Supplementary file 2. Table of primer sequences for RT-qPCR.

• Transparent reporting form

### Data availability

Sequencing data have been deposited in ArrayExpress under accession codes E-MTAB-10017 and E-MTAB-10018. All data generated or analysed during this study are included in the manuscript and supporting files; source data files have been provided for Figure 1, Figure 3, Figure 1-supplement figure 1 and Figure 3-supplement figure 1.

The following datasets were generated:

| Author(s) | Year | Dataset title | Dataset URL | Database and Identifier |
|---|---|---|---|---|
| Osnato A, Brown S, Vallier L, Ortmann D, Rugg-Gunn PJ | 2021 | TGFβ signalling is required to maintain pluripotency of naïve human pluripotent stem cells: ChIP-sequencing data | https://www.ebi.ac.uk/ar-rayexpress/experiments/E-MTAB-10017 | ArrayExpress, E-MTAB-10017 |
| Osnato A, Brown S, Vallier L, Ortmann D, Rugg-Gunn PJ | 2021 | TGFβ signalling is required to maintain pluripotency of naïve human pluripotent stem cells: scRNA-sequencing data | https://www.ebi.ac.uk/ar-rayexpress/experiments/E-MTAB-10018 | ArrayExpress, E-MTAB-10018 |

The following previously published datasets were used:

| Author(s) | Year | Dataset title | Dataset URL | Database and Identifier |
|---|---|---|---|---|
| Collier AJ | 2021 | Widespread reorganisation of pluripotent factor binding and gene regulatory interactions between human pluripotent | https://www.ncbi.nlm.nih.gov/geo/query/acc.cgi?acc=GSE133126 | NCBI Gene Expression Omnibus, GSE133126 |

| | | states | | |
|---|---|---|---|---|
| Collier AJ | 2017 | Comprehensive Cell Surface Protein Profiling Identifies Specific Markers of Human Naive and Primed Pluripotent States | https://www.ncbi.nlm.nih.gov/geo/query/acc.cgi | NCBI Gene Expression Omnibus, GSE93241 |
| Petropoulos S | 2016 | Single-Cell RNA-Seq Reveals Lineage and X Chromosome Dynamics in Human Preimplantation Embryos | https://www.ebi.ac.uk/arrayexpress/experiments/E-MTAB-3929/ | ArrayExpress, E-MTAB-3929 |
| Rostovskaya M | 2019 | Capacitation of human naïve pluripotent stem cells for multi-lineage differentiation | https://www.ncbi.nlm.nih.gov/geo/query/acc.cgi?acc=GSE123055 | NCBI Gene Expression Omnibus, GSE123055 |
| Wojdyla K | 2020 | Cell-Surface Proteomics Identifies Differences in Signaling and Adhesion Protein Expression between Naive and Primed Human Pluripotent Stem Cells | http://proteomecentral.proteomexchange.org/cgi/GetDataset?ID=PXD015359 | ProteomeXchange, PXD015359 |
| Xiang L | 2020 | A developmental landscape of 3D-cultured human pre-gastrulation embryos | https://www.ncbi.nlm.nih.gov/geo/query/acc.cgi/GSE136447 | NCBI Gene Expression Omnibus, GSE136447 |
| Pastor WA | 2018 | TFAP2C regulates transcription in human naive pluripotency by opening enhancers | https://www.ncbi.nlm.nih.gov/geo/query/acc.cgi?acc=GSE101074 | NCBI Gene Expression Omnibus, GSE101074 |
| Takashima Y | 2014 | Resetting Transcription Factor Control Circuitry Towards Ground State Pluripotency In Human | https://www.ncbi.nlm.nih.gov/geo/query/acc.cgi/GSE60945 | NCBI Gene Expression Omnibus, GSE60945 |
| Ji X | 2016 | 3D Chromosome Regulatory Landscape of Human Pluripotent Cells | https://www.ncbi.nlm.nih.gov/geo/query/acc.cgi/GSE69647 | NCBI Gene Expression Omnibus, GSE69647 |
| Theunissen TW | 2014 | Systematic Identification of Defined Conditions for Induction and Maintenance of Naive Human Pluripotency | https://www.ncbi.nlm.nih.gov/geo/query/acc.cgi/GSE59434 | NCBI Gene Expression Omnibus, GSE59434 |
| Theunissen TW | 2016 | Molecular Criteria for Defining the Naive Human Pluripotent State | https://www.ncbi.nlm.nih.gov/geo/query/acc.cgi/GSE75868 | NCBI Gene Expression Omnibus, GSE75868 |
| Gifford CA | 2013 | Transcriptional and Epigenetic Dynamics During Specification of Human Embryonic Stem Cells | https://www.ncbi.nlm.nih.gov/geo/query/acc.cgi/GSE46130 | NCBI Gene Expression Omnibus, GSE46130 |

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
