## [Decision Letter]

**Acceptance summary:**

The study provides important new insight into extrinsic factors regulating the naïve pluripotent state and trophoblast differentiation.

**Decision letter after peer review:**

Thank you for submitting your article "TGFβ signalling is required to maintain pluripotency of naïve human pluripotent stem cells" for consideration by *eLife*. Your article has been reviewed by 2 peer reviewers, and the evaluation has been overseen by a Reviewing Editor and Marianne Bronner as the Senior Editor. The following individual involved in review of your submission has agreed to reveal their identity: William Pastor (Reviewer #2).

Essential revisions:

1) Please provide further characterization of the trophoblast cells that you describe, particularly in light of two recent studies in Cell Stem Cell from the Smith and Takashima labs.

2) Please provide data on the capacity for somatic and germ line differentiation of these cells, or comment on this capacity.

*Reviewer #2:*

This paper contains two major, interrelated findings. The authors start by demonstrating that TGFB signaling occurs in both the naïve and primed pluripotent states. They next perform ChIP-seq for SMAD2/3 and demonstrate some overlapping and many distinct regulatory targets of SMAD2/3 in naïve and primed hESCs. They subsequently demonstrate, via an inhibitor time course experiment and an inducible knockdown experiment, that loss of TGFB signaling directly results in loss of expression of pluripotency genes.

This finding gives rise to the second part of the paper and perhaps the most striking observation: that inhibition of TGFB in naïve media conditions is sufficient to induce trophoblast differentiation of pluripotent stem cells. Furthermore, during this process the cells show gene expression changes (e.g. transient high expression of the CDX2) analogous to what is observed during trophoblast specification in the human embryo. This is not the first manuscript to convert pluripotent cells to trophoblast, nor the first to demonstrate that TGFB pathway inhibition is important for trophoblast cell survival. However, it is remarkable insofar as it shows that alteration of a single signalling pathway is sufficient to convert pluripotent cells to trophoblast. This has important ramifications for understanding the first steps of human embryonic development.

There are few critical places where additional validation is important:

1. Given the controversy surrounding various human trophoblast differentiation protocols (e.g. BAP) and the similar gene expression patterns in trophoblast and amnion, the authors must confirm that their cells are in fact trophoblast and not amnion. The markers the authors highlight in their analyses (e.g. CDX2, HAND1, GATA2, GATA3) are sufficient to distinguish trophoblast from neuroectoderm but not from amnion.

2. Figure 3d: Guo et al. 2020 Bioarchive and Io et al. 2020 Bioarchive identify suitable markers to distinguish trophoblast and amnion identity. To be clear, given the starting cells and pattern of CDX2 gain and loss, it is probable that Osnato and colleagues are indeed generating trophoblast, but it is a matter of due diligence to rule out amnion.

3. On a related note, the trophoblasts generated in this protocol are a little under-described. Do they continue dividing, do they show heterogeneity or differentiation etc.?

*Reviewer #3:*

In this work, Osnato et al. test the hypothesis that TGFβ/Activin/Nodal signaling is active in naïve hPSCs. The authors employ genome-wide and ssRNA-Seq analyses, as well as loss of function/signaling inhibition approaches in primed vs naïve hPSC lines to demonstrate convincingly that SMAD2/3 is bound near and potentially regulating the expression of both naïve-specifying and shared pluripotency-associated genes (e.g., NANOG). ssRNA-Seq analyses on naïve and primed hPSCs undergoing differentiation following TGFβ/Activin/Nodal signaling inhibition demonstrated differential lineage-specifying outcomes with primed hPSCs rapidly activating neuroectodermal specification, while naïve hPSCs activated extra-embryonic trophectodermal differentiation, thus suggesting a key regulatory role for SMAD2/3. These studies collectively reveal, in a convincing manner, that the TGFβ/Activin/Nodal signaling axis may potentially exist in a developmental continuum in both primed and naïve states, with different outcomes during either the pre and post implantation developmental stages. This is a finding of potentially high impact, if this differential signaling of this TGFb axis could be validated in native, in vivo-derived pre-implantation embryos, and if the naïve-reverting methods utilized to derive these naïve hPSC lines could be shown to have normal biological relevance.

These studies very convincingly demonstrate that naïve hPSC rapidly differentiate to the extra-embryonic trophectodermal lineage upon TGFb signaling inhibition. The molecular data is clear and compelling. However, there is a recurring theme with some naïve-reverting methodologies published in the literature, including the ones employed here by the authors that include MEK inhibition: A multitude of previous published works have now demonstrated a restricted and limited multi-lineage embryonic differentiation to mesoderm, ectodermal and endodermal lineages in naïve hPSC lines. In particular, Hochedlinger's group has suggested that epigenomic imprints in murine LIF2i-cultured as well as naïve hPSC naïve states that employ prolonged culture with the MEK inhibitor PD0325901 (eg as in the t2iLGo cocktail the authors utilize in these studies ) produce broadly defective and aberrantly hypomethylated PSC lines; thus explaining their abnormal embryonic differentiation capacities. Can the authors comment on these caveats of current naïve-reverting small molecule cocktails, and the capacity of the naïve hPSC used in their own studies to possess normal multi-lineage differentiation potency (eg to other lineages besides trophectoderm)?

Can the authors also discuss the role that TGFb signaling may play in normal multi-lineage differentiation in the naïve hPSC they have studied here? Do other signaling pathways initiate multi-lineage embryonic differentiation in naïve hPSC prepared with their methods?

---

## [Author Response]

Essential revisions:1) Please provide further characterization of the trophoblast cells that you describe, particularly in light of two recent studies in Cell Stem Cell from the Smith and Takashima labs.

Please see our response below.

2) Please provide data on the capacity for somatic and germ line differentiation of these cells, or comment on this capacity.

Please see our response below.

Reviewer #2:This paper contains two major, interrelated findings. The authors start by demonstrating that TGFB signaling occurs in both the naïve and primed pluripotent states. They next perform ChIP-seq for SMAD2/3 and demonstrate some overlapping and many distinct regulatory targets of SMAD2/3 in naïve and primed hESCs. They subsequently demonstrate, via an inhibitor time course experiment and an inducible knockdown experiment, that loss of TGFB signaling directly results in loss of expression of pluripotency genes.This finding gives rise to the second part of the paper and perhaps the most striking observation: that inhibition of TGFB in naïve media conditions is sufficient to induce trophoblast differentiation of pluripotent stem cells. Furthermore, during this process the cells show gene expression changes (e.g. transient high expression of the CDX2) analogous to what is observed during trophoblast specification in the human embryo. This is not the first manuscript to convert pluripotent cells to trophoblast, nor the first to demonstrate that TGFB pathway inhibition is important for trophoblast cell survival. However, it is remarkable insofar as it shows that alteration of a single signalling pathway is sufficient to convert pluripotent cells to trophoblast. This has important ramifications for understanding the first steps of human embryonic development.There are few critical places where additional validation is important:1. Given the controversy surrounding various human trophoblast differentiation protocols (e.g. BAP) and the similar gene expression patterns in trophoblast and amnion, the authors must confirm that their cells are in fact trophoblast and not amnion. The markers the authors highlight in their analyses (e.g. CDX2, HAND1, GATA2, GATA3) are sufficient to distinguish trophoblast from neuroectoderm but not from amnion.

We agree with the reviewer that this is an important point to clarify. In response, we have used three recent studies (Guo et al., 2021; Io et al., 2021; Zhao et al., 2021) to compile a set of genes reported in these papers to be expressed preferentially in amnion (including BAP cells) compared to trophoblast cells. We have then examined our 10X scRNA-seq timecourse dataset for the expression of these marker genes. The overall pattern is that most of the amnion-associated genes are not detectable in any of the scRNAseq clusters (Figure 4—figure supplement 1g). This includes marker genes such as *ISL1* and*CDH10* (Figure 4—figure supplement 1g and Author response image 1, B). Additionally, Io and colleagues reported that BAPs express HLA Class I whereas human trophectoderm / trophoblast do not, and accordingly we do not detect *HLA-A/B/C* transcripts in naïve hPSCs following SB treatment (Figure 4—figure supplement 1g and Author response image 1). Of note, some marker genes, such as *CTSV* and *TPM1*, are reported to be expressed in both amnion and trophoblast and, as expected, were upregulated in cluster E (corresponding to the trophoblast-like cell population) (Figure 4—figure supplement 1g and Author response image 1). Lastly, we found that genes reported to be expressed in trophoblast but not in amnion cells, such as *CCKBR* (Guo et al., 2021), are expressed in the vast majority of cells in cluster E (Author response image 1). Taken together, these results lead us to conclude that TGFβ inhibition of naïve hPSCs in the conditions used in our study does not promote the induction of reported amnion cell markers, and therefore reinforces the predominant trophoblast identity of the cells in our population.

**Author response image 1. sa2fig1:** UMAP visualisation of gene expression in naïve cells following TGFβ inhibition, as measured by 10X scRNA-seq. a) Overview of the louvain clusters described in the manuscript. The plots show that reported b) amnion markers such as *ISL1* and *CDH10* (Guo et al., 2021) and c) BAP markers such as *HLA-A/B/C* (Io et al., 2021) are largely undetectable in this time course data set. d) *TMP1* and *CTSV* are reported to be expressed in both trophoblast and amnion cells, and e) *CCKBR* is reported to be more abundant in trophoblast compared to amnion cells (Guo et al., 2021).

2. Figure 3d: Guo et al. 2020 Bioarchive and Io et al. 2020 Bioarchive identify suitable markers to distinguish trophoblast and amnion identity. To be clear, given the starting cells and pattern of CDX2 gain and loss, it is probable that Osnato and colleagues are indeed generating trophoblast, but it is a matter of due diligence to rule out amnion.

Please see above.

3. On a related note, the trophoblasts generated in this protocol are a little under-described. Do they continue dividing, do they show heterogeneity or differentiation etc.?

We thank the reviewer for encouraging us to expand on this point, which we agree was underdeveloped in our original submission.

The naïve hPSCs treated with SB for 5 and 7 days are a heterogeneous cell population, as can be seen by the presence of distinct cell types (clusters) in the 10X scRNA-seq analysis (Figure 4b) and immunofluorescence microscopy images (Figure 3g and Figure 3—figure supplement 1f, g). Within the trophoblast-like cell subpopulation (cluster E), however, the cells’ transcriptional profiles appear to be relatively uniform and we detect very few cells that express markers of differentiated trophoblast cell types (syncytiotrophoblast and extravillous trophoblast; Author response image 2). However, it was unknown if the trophoblast-like cells in these cultures could a) give rise to proliferative trophoblast stem cells (TSCs) and b) if they could differentiate into specialised trophoblast cell types. We have now addressed these two additional points in the revised manuscript and we show the new data in Figures 3h and 3i.

In this new set of experiments, we cultured naïve hPSCs in the presence of SB for 5 days and then transferred the cells into TSC media (Dong et al., 2020; Okae et al., 2018). Following exposure to TSC conditions the cells rapidly and uniformly acquired a homogeneous TSC-like morphology. The cells expressed TSC markers, such as GATA3 and CK19 (Figure 3h) and *CK7*, *ITGA6* and *TP63* (Figure 3i), and could be passaged and maintained in these conditions with stable growth and morphology. To examine whether the cells can differentiate into specialised trophoblast derivatives, we induced the naïve hPSCderived TSCs to differentiate by switching the cells to STB and EVT media (Dong et al., 2020; Okae et al., 2018). This led to the downregulation of TSC genes and the upregulation of STB and EVT markers, such as SDC1 and HLA-G, respectively (Figure 3h, i). We therefore conclude that blocking TGFβ signalling in naïve hPSCs rapidly destabilises the pluripotency network and allows the cells to undergo differentiation towards trophoblast-like cells, and this includes cells that can give rise to multipotent and proliferative TSCs.

**Author response image 2. sa2fig2:** UMAP visualisation of gene expression in naïve cells following TGFβ inhibition, as measured by 10X scRNA-seq. The plots show that trophoblast markers such as *GATA3* are uniformly expressed in ‘Cluster E’, but markers of syncytiotrophoblast (STB) such as *SDC1* and extravillous trophoblast (EVT) such as *MMP2* are expressed in only a small number of cells in this population (indicated by the arrows).

Reviewer #3:In this work, Osnato et al. test the hypothesis that TGFβ/Activin/Nodal signaling is active in naïve hPSCs. The authors employ genome-wide and ssRNA-Seq analyses, as well as loss of function/signaling inhibition approaches in primed vs naïve hPSC lines to demonstrate convincingly that SMAD2/3 is bound near and potentially regulating the expression of both naïve-specifying and shared pluripotency-associated genes (e.g., NANOG). ssRNA-Seq analyses on naïve and primed hPSCs undergoing differentiation following TGFβ/Activin/Nodal signaling inhibition demonstrated differential lineage-specifying outcomes with primed hPSCs rapidly activating neuroectodermal specification, while naïve hPSCs activated extra-embryonic trophectodermal differentiation, thus suggesting a key regulatory role for SMAD2/3. These studies collectively reveal, in a convincing manner, that the TGFβ/Activin/Nodal signaling axis may potentially exist in a developmental continuum in both primed and naïve states, with different outcomes during either the pre and post implantation developmental stages. This is a finding of potentially high impact, if this differential signaling of this TGFb axis could be validated in native, in vivo-derived pre-implantation embryos, and if the naïve-reverting methods utilized to derive these naïve hPSC lines could be shown to have normal biological relevance.These studies very convincingly demonstrate that naïve hPSC rapidly differentiate to the extra-embryonic trophectodermal lineage upon TGFb signaling inhibition. The molecular data is clear and compelling. However, there is a recurring theme with some naïve-reverting methodologies published in the literature, including the ones employed here by the authors that include MEK inhibition: A multitude of previous published works have now demonstrated a restricted and limited multi-lineage embryonic differentiation to mesoderm, ectodermal and endodermal lineages in naïve hPSC lines. In particular, Hochedlinger's group has suggested that epigenomic imprints in murine LIF2i-cultured as well as naïve hPSC naïve states that employ prolonged culture with the MEK inhibitor PD0325901 (eg as in the t2iLGo cocktail the authors utilize in these studies ) produce broadly defective and aberrantly hypomethylated PSC lines; thus explaining their abnormal embryonic differentiation capacities. Can the authors comment on these caveats of current naïve-reverting small molecule cocktails, and the capacity of the naïve hPSC used in their own studies to possess normal multi-lineage differentiation potency (eg to other lineages besides trophectoderm)?Can the authors also discuss the role that TGFb signaling may play in normal multi-lineage differentiation in the naïve hPSC they have studied here? Do other signaling pathways initiate multi-lineage embryonic differentiation in naïve hPSC prepared with their methods?

The reviewer raises many interesting points about the nature of naïve hPSCs. We respond to the comments below and we have also added new text to the discussion to incorporate these helpful considerations (page 16).

In our current study, we have used two conditions to maintain naïve hPSC lines: t2iLGö and PXGL (Bredenkamp et al., 2019; Rostovskaya et al., 2019; Takashima et al., 2014). Many previous publications have shown that cells maintained in these two conditions display key hallmarks of primate naïve pluripotency (reviewed in Boroviak and Nichols, 2017; Dong et al., 2019; Semi and Takashima, 2021). In particular, the cells are transcriptionally similar to human and primate preimplantation epiblast cells including the expression of primate-specific factors and transposable elements; female cells have two active X-chromosomes and express *XIST* in keeping with human preimplantation epiblast cells; the cells are DNA hypomethylated; and their main energy production pathway is oxidative phosphorylation. Importantly, naïve hPSCs also have the ability to differentiate into trophoblast and hypoblast lineages (Bayerl et al., 2021; Castel et al., 2020; Cinkornpumin et al., 2020; Dong et al., 2020; Guo et al., 2021; Io et al., 2021; Linneberg-Agerholm et al., 2019). This is most strikingly demonstrated by the ability of naïve hPSC to aggregate into ‘blastoids’ that form and self-organise the three early lineages (epiblast, trophoblast and hypoblast) (Liu et al., 2021; Yanagida et al., 2021; Yu et al., 2021). The ability of naïve hPSC to generate extraembryonic cell types with high efficiency, in contrast to naïve mouse PSCs which do so with very low efficiency, is probably due to species differences in the timing and allocation of early cell lineages. In humans, the three lineages are not transcriptionally distinct until quite late in development when the blastocyst begins to expand and cavitate (Blakeley et al., 2015; Meistermann et al., 2021; Petropoulos et al., 2016). Furthermore, isolated inner cell masses from human blastocysts have the capability to generate trophoblast cells, suggesting that the capacity for undifferentiated cells in the human blastocyst to generate extraembryonic cell types is an intrinsic property and one that is retained in naïve hPSCs in culture (Guo et al., 2021).

Because naïve hPSCs have preimplantation epiblast identity, they cannot respond directly to signals that induce postimplantation germ layer specification. It would be unusual and unexpected if they could as this would represent a departure from their developmental identity. In human embryos, it takes approximately seven days for epiblast cells to transition from preimplantation to postimplantation just prior to gastrulation (considerably longer compared to the mouse). This transition can be reproduced in vitro through a process termed capacitation, or formative transition (Rostovskaya et al., 2019). Here, naïve hPSCs (cultured in t2iLGö and PXGL) are transitioned into a pluripotent state similar to primed hPSCs and these ‘capacitated’ cells can respond directly to signalling cues and undergo multilineage germ layer induction (Rostovskaya et al., 2019). Remarkably, the timing of in vitro capacitation (~8 days) is very similar to the timing in vivo. Taken together, our prior work and other studies have therefore shown that naïve hPSCs grown in t2iLGö and PXGL have the capacity to generate trophoblast and hypoblast cells but not directly ectoderm, mesoderm and endoderm until they have undergone capacitation, after which they can efficiently make the major germ layer lineages. We have expanded the discussion to highlight the differences in developmental identity and potential between naïve and primed hPSCs and how this might impact their different responses to TGFβ inhibition.

Regarding the signalling pathways, capacitation of naïve hPSCs is achieved by maintaining Wnt inhibition (using the XAV939 tankyrase inhibitor) and by removing LIF and PD0325901 (MEK inhibitor) from the conditions (Rostovskaya et al., 2019). We predict that a source of TGFβ/Activin/Nodal is provided by the substrate (typically Geltrex or Matrigel) and also produced endogenously by the cells. It is currently unknown whether inhibiting TGFβ/Activin/Nodal compromises formative transition, but based on our current and prior work we predict that it would be. Once the naïve hPSCs have been capacitated for ~7 days, the cells can be stabilised in a primed-like state using media containing XAV939, Activin and FGF2 (XAF media). Day 7 capacitated cells and cells transferred to XAF media can undergo germ layer induction using protocols optimised for primed hPSC, and with similar differentiation efficiencies (Rostovskaya et al., 2019).

Lastly, the comment about the inclusion of the MEK inhibitor PD0325901 in t2iLGö and PXGL (and most other mouse and human 2i-based methods) is a good one. It is clear that culture methods are suboptimal and indeed the inclusion of PD0325901 has been linked to prolonged DNA hypomethylation and erosion of some imprint control regions (we see this too in our cultures). It is unlikely in our opinion however that the loss of certain imprints is responsible for the inability of naïve hPSCs to differentiate into the three germ layers (as we think the reviewer is querying). We base our opinion on several findings: (i) naïve hPSCs that have been derived and cultured in PD0325901-containing media can undergo efficient germ layer differentiation following capacitation. It is unclear at the moment whether the misregulation of imprinted genes might bias or limit the resultant capacitated cells – this important point remains to be examined in depth; (ii) parthenogenetic and androgenetic primed hPSC, which have very severe imprint dysregulation, are not globally defective in germ layer differentiation (Sagi et al., 2019; Stelzer et al., 2015). These cells can differentiate into ectoderm, mesoderm and endoderm derivatives with efficiencies that are not too different from normal biparental hPSCs, albeit with some tissue specific biases. We therefore suggest that the potential for naïve hPSCs to differentiate into extraembryonic cell types, and not directly into germ layer lineages, is less to do with imprint misregulation and more to do with their developmental identity that matches expectations based on the human blastocyst. Despite this, it would certainly be desirable to remove the MEK inhibitor PD0325901 from mouse and human naïve PSC cultures, and recent papers have made excellent progress in this area in mouse (Choi et al., 2017; Yagi et al., 2017) and human (Bayerl et al., 2021; Di Stefano et al., 2018; Khan et al., 2021).

References:

Bayerl et al. 2021. Principles of signaling pathway modulation for enhancing human naïve pluripotency induction. Cell Stem Cell. doi:10.1016/j.stem.2021.04.001

Blakeley et al. 2015. Defining the three cell lineages of the human blastocyst by single-cell RNA-seq. Development 142:3151–3165.

Boroviak T, Nichols J. 2017. Primate embryogenesis predicts the hallmarks of human naïve pluripotency. Development 144:175–186.

Bredenkamp et al. 2019. Wnt Inhibition Facilitates RNA-Mediated Reprogramming of Human Somatic Cells to naïve Pluripotency. Stem Cell Reports 13:1083–1098.

Castel et al. 2020. Induction of Human Trophoblast Stem Cells from Somatic Cells and Pluripotent Stem Cells. Cell Rep 33:108419.

Choi et al. 2017. Prolonged Mek1/2 suppression impairs the developmental potential of embryonic stem cells. Nature 548:219–223.

Cinkornpumin et al. 2020. Naïve Human Embryonic Stem Cells Can Give Rise to Cells with a Trophoblast like Transcriptome and Methylome. Stem Cell Reports 15:198–213.

Di Stefano et al. 2018. Reduced MEK inhibition preserves genomic stability in naïve human embryonic stem cells. Nat Methods 15:732–740.

Dong et al. 2020. Derivation of trophoblast stem cells from naïve human pluripotent stem cells. *eLife* 9.

doi:10.7554/*eLife*.52504

Dong et al. 2019. Recent insights into the naïve state of human pluripotency and its applications. Exp Cell Res 385:111645.

Guo et al. Human naïve epiblast cells possess unrestricted lineage potential. Cell Stem Cell 28:1040 1056.e6.

Io et al. 2021. Capturing human trophoblast development with naïve pluripotent stem cells in vitro. Cell Stem Cell 28:1023–1039.e13.

Khan et al. 2021. Probing the signaling requirements for naïve human pluripotency by high-throughput chemical screening. Cell Rep 35:109233.

Linneberg-Agerholm et al. 2019. Naïve human pluripotent stem cells respond to Wnt, Nodal and LIF signalling to produce expandable naïve extra-embryonic endoderm. Development 146.

doi:10.1242/dev.180620

Liu et al. 2021. Modelling human blastocysts by reprogramming fibroblasts into iBlastoids. Nature 591:627–632.

Meistermann et al. 2021. Integrated pseudotime analysis of human pre-implantation embryo single-cell transcriptomes reveals the dynamics of lineage specification. Cell Stem Cell doi:10.1016/j.stem.2021.04.027

Okae et al. 2018. Derivation of Human Trophoblast Stem Cells. Cell Stem Cell 22:50–63.

Petropoulos et al. 2016. Single-Cell RNA-Seq Reveals Lineage and X Chromosome Dynamics in Human Preimplantation Embryos. Cell 165:1012-1026.

Rostovskaya et al. 2019. Capacitation of human naïve pluripotent stem cells for multi-lineage differentiation. Development 146. doi:10.1242/dev.172916

Sagi et al. 2019. Distinct Imprinting Signatures and Biased Differentiation of Human Androgenetic and Parthenogenetic Embryonic Stem Cells. Cell Stem Cell 25:419–432.

Semi and Takashima. 2021. Pluripotent stem cells for the study of early human embryology. Dev Growth Differ 63:104–115.

Stelzer et al. 2015. Differentiation of human parthenogenetic pluripotent stem cells reveals multiple tissue- and isoform-specific imprinted transcripts. Cell Rep 11:308–320.

Takashima et al. 2014. Resetting transcription factor control circuitry toward ground-state pluripotency in human. Cell 158:1254–1269.

Yagi et al. 2017. Derivation of ground-state female ES cells maintaining gamete-derived DNA methylation. Nature 548:224–227.

Yanagida et al. 2021. naïve stem cell blastocyst model captures human embryo lineage segregation. Cell Stem Cell 28:1016–1022.

Yu et al. 2021. Blastocyst-like structures generated from human pluripotent stem cells. Nature 591:620 –626.

Zhao et al. 2021. Reprogrammed iBlastoids contain amnion-like cells but not trophectoderm. bioRxiv doi:10.1101/2021.05.07.442980